# LINE-1 regulates cortical development by acting as long non-coding RNAs

Damiano Mangoni [1], Alessandro Simi [1], Pierre Lau[1], Alexandros Armaos[1], Federico Ansaloni[1], Azzurra Codino [1], Devid Damiani[1], Lavinia Floreani[2], Valerio Di Carlo [1], Diego Vozzi, Francesca Persichetti[3], Claudio Santoro[3], Luca Pandolfini [1], Gian Gaetano Tartaglia [1], Remo Sanges[1,2] ✉ & Stefano Gustincich [1] ✉

Long Interspersed Nuclear Elements-1s (L1s) are transposable elements that constitute most of the genome's transcriptional output yet have still largely unknown functions. Here we show that L1s are required for proper mouse brain corticogenesis operating as regulatory long non-coding RNAs. They contribute to the regulation of the balance between neuronal progenitors and differentiation, the migration of post-mitotic neurons and the proportions of different cell types. In cortical cultured neurons, L1 RNAs are mainly associated to chromatin and interact with the Polycomb Repressive Complex 2 (PRC2) protein subunits *enhancer of Zeste homolog 2* (Ezh2) and *suppressor of zeste 12* (Suz12). L1 RNA silencing influences PRC2's ability to bind a portion of its targets and the deposition of tri-methylated histone H3 (H3K27me3) marks. Our results position L1 RNAs as crucial signalling hubs for genome-wide chromatin remodelling, enabling the fine-tuning of gene expression during brain development and evolution.

19% of the mouse genome is made up by more than 100,000 Long Interspersed Nuclear Elements 1 (L1s)[1,2]. The majority of transcribed L1s are of the most recently evolved families L1MdA, L1MdGf and L1MdTf, defined by a variable region in the 5'UTR[3]. Although less than 10% of L1s are full-length, these are potentially capable of autonomous retro-transposition, representing a threat to genome stability[4]. Cells have developed several mechanisms to repress L1 expression, including DNA methylation, histone modifications and affecting stability of L1 transcripts through the activity of MIWI/piRNA pathway and RNA helicases[5,6]. Nevertheless, L1 RNAs are expressed in a controlled pattern in time and space[7]. In fact, emerging evidence suggests that they can function as regulatory long non-coding RNAs (lncRNAs), controlling transcriptional and chromatin landscapes, and are required for mouse embryonic stem cell (mESC) self-renewal and pre-implantation development[8,9]. Somatic L1 retrotransposition has been observed in the neuronal lineage[10,11]. Uncontrolled expression of L1s triggers

oxidative stress-induced DNA strand breaks leading to cell death and neurodegeneration[12].

The mammalian neocortex develops into a six-layered structure comprising hundreds of neuronal subtypes[13]. These are generated from neural progenitors in a sequential manner, with subtypes located in the deepest layer generated at the earliest stage. Neural progenitors change their potential over time, thereby determining the subtype of newly generated neurons[14].

Starting from the observations that L1s are active in neurons and they function as lncRNAs in ESC self-renewal, we hypothesized that L1s may operate as regulatory lncRNAs in neuronal differentiation.

## Results

### L1 silencing alters cortical neurogenesis in vivo

The expression of L1MdA, L1MdGf and L1MdTf RNAs was characterized during mouse brain corticogenesis. As shown by reverse-transcriptase

[1]Central RNA Laboratory, Istituto Italiano di Tecnologia (IIT), Genova, Italy. [2]Area of Neuroscience, International School for Advanced Studies (SISSA), Trieste, Italy. [3]Department of Health Sciences and Research Center on Autoimmune and Allergic Diseases (CAAD), University of Piemonte Orientale (UPO), Novara, Italy. ✉e-mail: remo.sanges@gmail.com; stefano.gustincich@iit.it

quantitative PCR (RT-qPCR), all L1 RNAs were induced during cortical development, up to 4-fold (Fig. S1A). To investigate their function, we studied the effects of L1 RNAs silencing by in utero electroporation (i.u.e.) at E12.5 using a plasmid expressing a short hairpin RNA sequence directed against a conserved region of L1s (shL1-a) and collecting cortical samples at E13.5, E14.5 and E18.5 (Fig. 1a). The ability of shL1-a to silence L1 RNAs was verified by RT-qPCR comparing L1 transcripts levels of FACS-sorted cells co-expressing GFP (GFP$^+$) derived from shL1-a and shCtrl i.u.e. cortices at E14.5 (60%, 56% and 54% of knockdown efficiency for L1MdA, L1MdGf and L1MdTf, respectively; Fig. 1b). We examined the effect of shL1-a versus shCtrl on neuronal progenitors and post-mitotic neurons by immunohistochemistry (IHC) for cell type-specific markers.

In the L1-silenced area, Pax6-expressing (Pax6$^+$) radial glial progenitor cells in the ventricular zone (VZ) were significantly fewer at E13.5 ($P = 0.0016$) and E14.5 ($P = 5.209e-5$), whereas at the same developmental stages, cells expressing the pro-neuronal gene marker NeuroD1 were more numerous (E13.5 $P = 0.0153$; E14.5 $P = 0.0045$; Fig. 1c–f). The decrease of Pax6$^+$ ($P = 0.0017$) and the increase of NeuroD1$^+$ ($P = 0.0011$) cells at E14.5 were reproduced when transfecting the cortex with another previously validated shRNA directed against L1s (shL1-b; Fig. S1B–C)[9]. L1 RNAs silencing at E14.5 also increased the number of cells positive for the neurogenic marker NeuroD2 and decreased the number of Tbr2$^+$ intermediate progenitors in the neocortex ($P = 0.0101$ and $P = 0.0325$, respectively; Fig. S1D–E). Interestingly, Tbr2$^+$ cells accumulated in the VZ and SVZ ($P = 0.0029$; Fig. S1F). Among post-mitotic neurons, cells expressing the transcription factors Tbr1 and Ctip2, markers of deep-layer neurons, were reduced at E14.5 ($P = 0.0030$ and $P = 0.0025$, respectively; Fig. S1G–H). At E18.5, the proportions of Tbr1 and Ctip2 cells were restored, Cux1$^+$ upper-layer

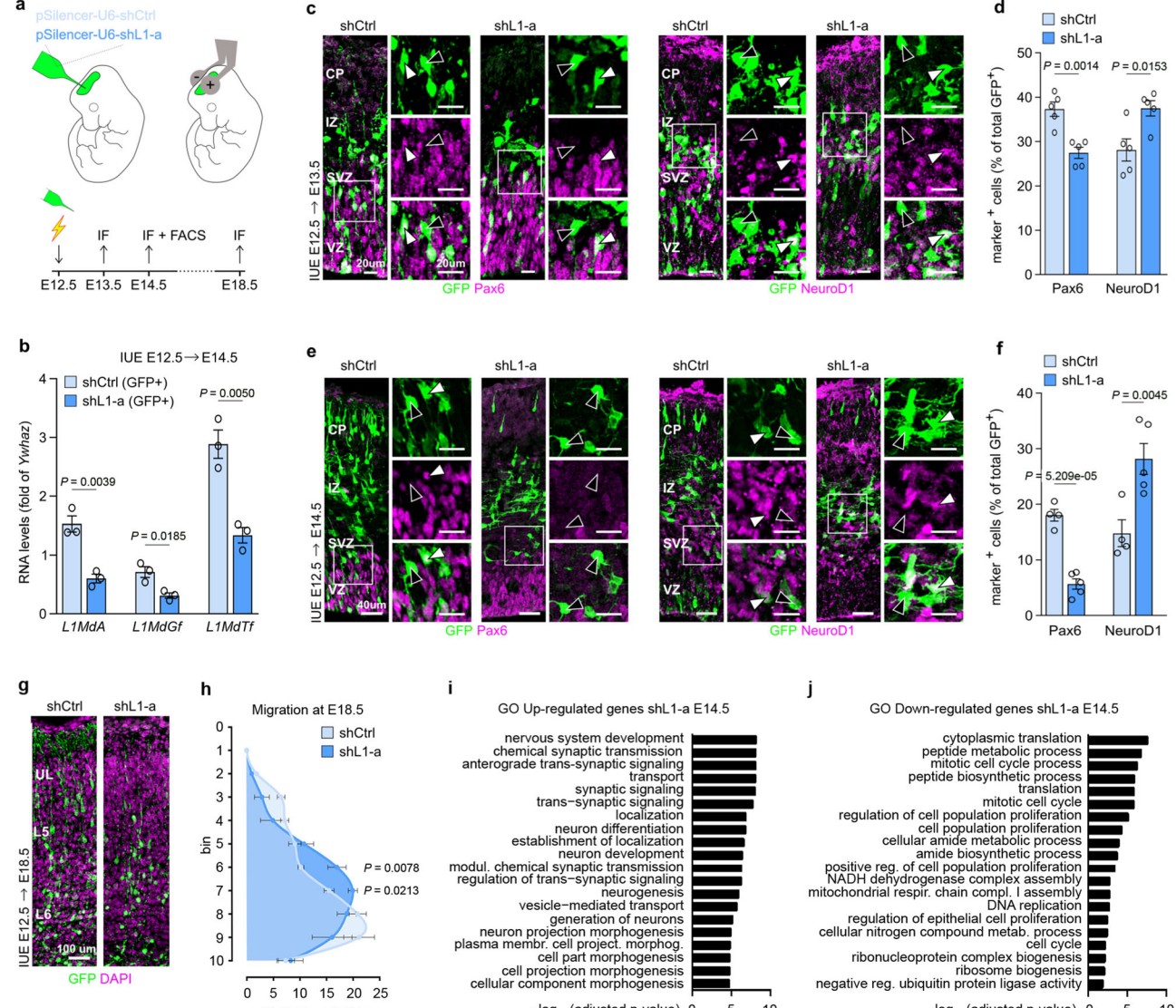

**Fig. 1 | L1 silencing alters neocortical development. a** Schematic of in utero electroporation (i.u.e.) procedure (up) and experimental timeline (bottom). **b** Expression of L1 transcripts in GFP$^+$ cells from E14.5 cortices i.u.e with shCtrl or shL1-a. RNA levels are normalized on shCtrl group. Data are mean ± s.e.m. $n = 3$ technical replicates. Two-sided unpaired $t$-test. **c**, **e** Immunofluorescence staining for neuronal progenitor (Pax6) and pro-neuronal (NeuroD1) markers at E13.5 (**c**) and E14.5 (**e**). **d**, **f** Quantification of GFP$^+$/marker$^+$ cells at E13.5 (**d**) and E14.5 (**f**). Data are mean ± s.e.m.; E13.5: $n = 5$ each; E14.5: $n = 4$ shCtrl, $n = 5$ shL1-a. Two-sided unpaired $t$-test. **g** Immunofluorescence staining for DAPI and GFP at E18.5. **h** Radial distribution of GFP$^+$ cells at E18.5. Data are mean ± s.e.m.; $n = 4$ shCtrl, $n = 3$ shL1-a. Multiple $t$-test with correction for multiple comparison with Sidak-Bonferroni's test. **i**, **j** Top GO terms under the biological process category for up-regulated (**i**) and down-regulated (**j**) genes in mouse GFP$^+$ cells after i.u.e. of shL1-a versus shCtrl. $p$ values were determined by gprofiler2 using a default hypergeometric test and correction for multiple testing has been performed by the g:SCS algorithm. Source data are provided as a Source Data file.

callosal neurons were more than doubled ($P = 0.0086$), while those expressing the marker Satb2[+], another upper-layer callosal marker, were similar to controls (Fig. S1I–J). Furthermore, cells expressing shL1-a showed impaired migration throughout the cortical layers, with most accumulating in the deeper layers of the cortical plate and only a few reaching the outer layers of the neocortex (bin 6: $P = 0.0078$, bin 7: $P = 0.0213$; Fig. 1g–h).

These results suggest that L1 RNAs have a dual effect on neocortical development: (1) they limit neuronal commitment of progenitor cells towards the upper layer neuronal fate, maintaining the correct pool of radial glial cells in the VZ, and (2) they play a role in the ability of newborn neurons to correctly migrate along the cortical plate.

To identify the molecular pathways underpinning these roles, we performed RNA sequencing (RNA-Seq) on FACS-sorted E14.5 GFP[+] cells from shL1-a and shCtrl i.u.e. cortices. L1 silencing upregulated 3480 genes and downregulated 2705 genes (Supplementary Data 2). Transposable elements expression analysis using TEspeX[15] showed changes of expression in L1 families transcribed in the developing mouse brain cortex, particularly for those most evolutionarily recent, consistent with RT-qPCR data (Fig. S1K). Among the upregulated genes, the transcription factors *Neurod1, Neurod2, Dlx1, Nfia, Nfib, Nfix, Emx1* and *Cux1* are involved in numerous neurodevelopmental processes, *Cntnap2, Ncam* and several *Pcdh* transcripts control neuronal migration and cell-to-cell interactions, while 5 subunits of GABA and 7 of ionotropic glutamate receptors as well as 2 sodium, 5 calcium and 21 potassium voltage-gated channels are involved in synaptic transmission and neuronal excitability. Gene Ontology (GO) analysis revealed a significant enrichment for terms related to nervous system development, axogenesis, and projection development. In contrast, downregulated genes included transcription factors *Ascl1, Neurog1, Neurog2, Neurod4* and Notch signalling components (*Notch1, Notch2, Notch4, Hes1* and *Hes5*). In addition, 10 *Atp5* subunits, 21 *Cox* genes, 26 *Nduf* subunits, 43 mitochondrial ribosomal proteins, 4 *Mterf* genes, *Mtch2, Mpc2, Mtfp1* and *Mtfr2* were all decreased, suggesting an inhibitory effect on the expression of nuclear-encoded mitochondrial genes and of those involved in mitochondria maintenance. This gene set was enriched for terms related to cell metabolism such as protein synthesis, mitochondrial activity, ribosome assembly and cell division (Supplementary Data 2, Fig. 1i, j). Comparing the list of differentially expressed genes (DEGs) with those altered upon L1 RNAs silencing in mESCs[9] indicated 215 downregulated genes in common ($P = 5.1e-17$, Fisher's exact test, Jaccard index = 0.1) and 588 shared upregulated genes ($P = 0.059$, Fisher's exact test, Jaccard index = 0.1). The 215 downregulated genes are shown in Supplementary Data 2 and included translation factors (*Eef1e1, Eef1g, Eif1ax, Eif2s2, Eif3e, Eif3h, Eif3i, Eif3m* and *Etf1*) and ribosomal proteins for large (*Rpl13a, Rpl14, Rpl22, Rpl35a, Rpl37, Rpl37a, Rpl38, Rpl6, Rpl7* and *Rpl7l1*) and small subunits (*Rps12, Rps14, Rps15a, Rps19, Rps21, Rps24, Rps25, Rps27a, Rps3, Rps3a3, Rps7* and *Rps8*), with GO terms enriched for ribosome biogenesis and translation (Supplementary Data 2, Fig. S1L). In mESCs, L1 RNAs expression regulates cell proliferation by inducing rRNA synthesis and ribosomal biogenesis, enabling rapid growth of the early embryo. These results suggest L1 RNAs work through a similar mechanism to balance proliferation and differentiation in neuronal progenitors, contributing to cell maintenance in a growing state.

## L1 silencing impairs cortical cell development and maturation in vitro

To further study L1 RNAs role in neuronal differentiation, we isolated cells from E17.5 embryonic cerebral cortex and cultured them for 21 days in vitro (div). This is a well-characterized, widely used primary neuronal cell culture system to model later stages of corticogenesis. Transcript levels for L1 subfamilies L1MdA, L1MdGf and L1MdTf increased progressively during the transition from early (3 div) to late maturation stages, reaching a 2-fold upregulation at 21 div (Fig. S2A).

Increased L1 RNAs expression correlated with de-repression of L1 promoters, sustained by decreased deposition of H3K9me3 (Fig. S2B) and lower methylation levels of the L1MdTf promoters (Fig. S2C–D)[16]. These changes were concurrent to a progressive decline of DNA methyltransferase 1 and 3b expression (Fig. S2E). Neuronal maturation in vitro is thus associated with increased expression of L1 RNAs at least in part due to epigenetic regulation of their promoters.

To explore the role of L1 RNAs in this in vitro model of neuronal maturation, cortical cultures were transduced 5 days after isolation with adeno-associated virus (AAV) expressing either shL1-a or shL1-b shRNA sequences, both targeting the conserved region of L1 Orf2, and then examined at 21 div (Fig. 2a, b). RNA-Seq analysis by TEspeX showed that both shL1s significantly decreased L1 transcripts levels, particularly those of the most evolutionarily recent L1 elements (Fig. S3A–B). RT-qPCR confirmed the reduction of L1MdA, L1MdGf, and L1MdTf subfamilies and on the conserved sequence of L1 Orf2. shL1-a showed a L1 knockdown efficiency of 40–45% and shL1-b of 35–40%, compared to non-infected (n.i.) cells or cells infected with a control shRNA (shCtrl; Fig. 2c). Many factors may negatively affect the efficiency of the knockdown exerted by shL1s including localization of L1 RNAs, formation of secondary structures, interactions with DNA and proteins that could mask the sequences targeted by the shRNAs, and the high heterogeneity of transcribed L1 elements which gives rise to different populations of L1 transcripts in terms of RNA sequences, some of them resistant even to shRNA designed on highly conserved regions.

The shL1-a dramatically changed cultured cell's transcriptional profile, with 3968 upregulated genes and 3933 downregulated genes when compared to the shCtrl (FDR $< = 0.05$, limma voom test). A similar observation was made using shL1-b, with 4758 upregulated genes and 2958 downregulated genes when compared to the shCtrl (FDR $< = 0.05$, limma voom test). The Jaccard similarity coefficient between shL1-a and shL1-b was 0.5 and 0.4 for the upregulated and downregulated gene sets, respectively. The statistical testing of the overlapping genes was significant for the common upregulated genes ($n = 2710$, $P = 0$, Fisher's exact test) and the common downregulated genes ($n = 2038$, $P = 0$, Fisher's exact test). Therefore, we combined the results for the two shL1s and a more stringent filtering was applied by intersecting the consistently dysregulated genes obtained when comparing the two shL1s versus shCtrl and also versus the n.i. cells. This led to 3028 common DEGs (1606 upregulated and 1422 downregulated). There were no significant differences between shCtrl cells and n.i. cells (Supplementary Data 3, Figs. S3C, 2d). We then compared the DEGs induced by L1 knockdown with the genes that changed during the physiological maturation of non-infected cells from 5 to 21 div (Supplementary Data 3, Fig. 2e). Genes upregulated during neuronal maturation were downregulated by shL1s and conversely, developmentally downregulated genes were upregulated by L1 silencing (shL1-a: Fig. 2e, top; shL1-b: Fig. 2e, bottom). Among genes down-regulated by shL1s, no predicted off-targets with no mismatches have been found for shL1a while *Fech* (ferrochelatase) is the only predicted off-target with no mismatches for shL1-b. *Kcnq1ot1* (KCNQ1 overlapping transcript 1) is the only predicted off-target with 1 mismatch with both shRNAs (Supplementary Data 4). As a note of caution, *Kcnq1ot1* is an antisense lncRNA recently shown to regulate chromatin status by recruiting chromatin-modifying complexes[17].

Upon L1 RNAs silencing, in vitro upregulated mRNAs included those that encode ribonucleoproteins (*Hnrnpa1, Hnrnpa3, Hnrnpa2b1, Hnrnpc, Hnrnpf, Hnrnpk, Hnrnpl, Hnrnpm, Hnrnpr*, among others), proteins involved in RNA processing (*Cmtr2, Dicer1, Drosha, Igf2bp2* and *Igf2bp3, Ythdf2*) and splicing (*Srsf1, Srsf2, Srsf3, Srsf4, Srsf7, Srsf11; Sf3a1, Sf3a2 Sf3b2, Sf3b3; U2af1, U2af2*, among others). They also include proteins that modify DNA (Dnmt1) and histones (*Ehmt1, Ehmt2, Epc1, Epc2, Hdac2, Hdac5, Hdac6, Hdac9, Kat5, Kat6a, Kat6b, Kdm1a,*

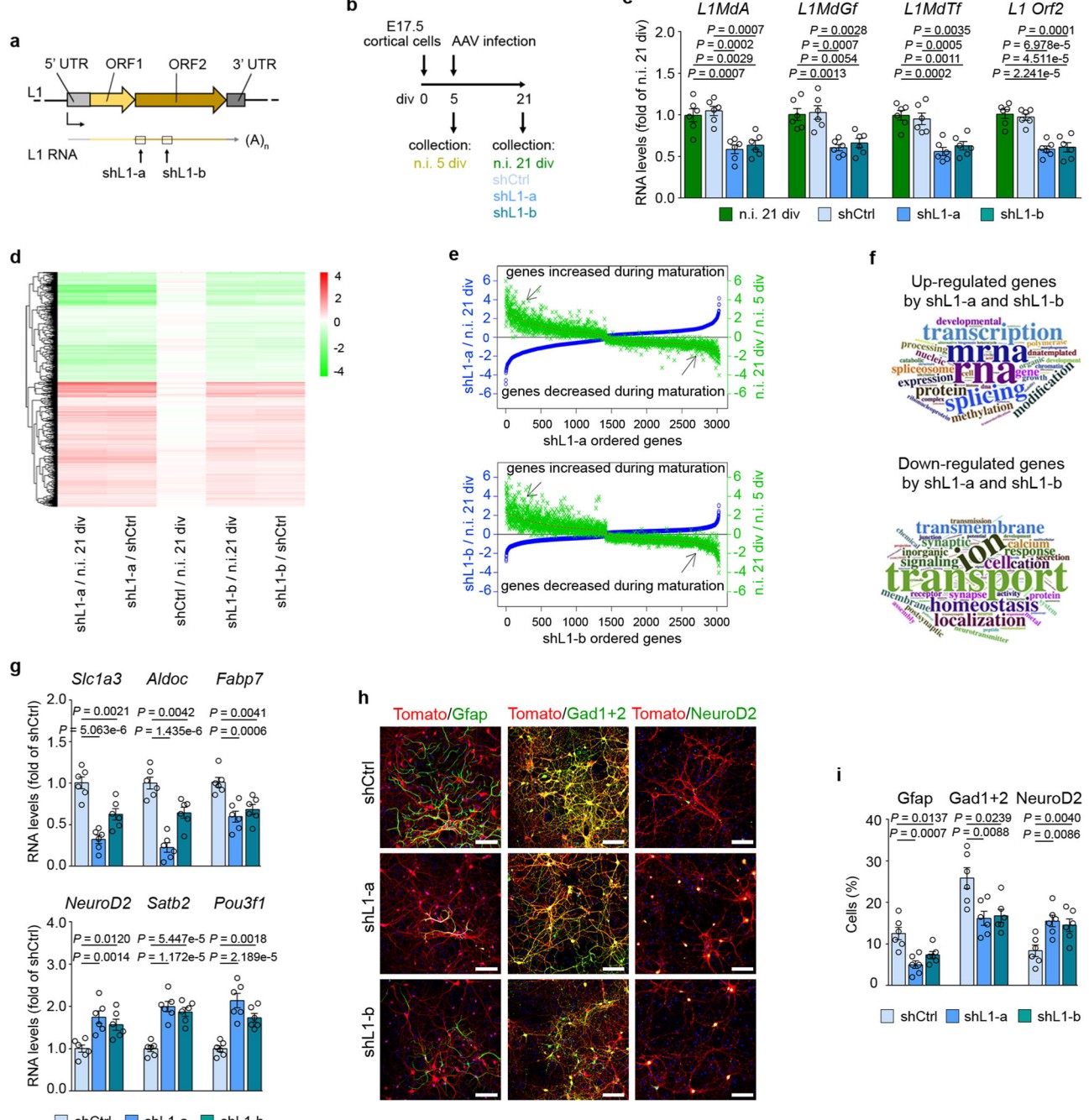

**Fig. 2 | L1 silencing impairs neuronal maturation. a** Schematic of the localization of shL1-a and shL1-b targeting L1 transcripts. **b** Schematic of the timeline of cortical cells infection. **c** Expression of L1 transcripts in cells infected with shCtrl, shL1-a and shL1-b. RNA levels are normalized on the non-infected (n.i.) group. **d** Heatmap of the log2 fold changes of DEGs in cells infected with shL1-a and shL1-b compared to n.i. or shCtrl. The rows correspond to the significant genes and the columns to the group comparisons. **e** Correlation analysis of DEGs in shL1-a (top) and shL1-b (bottom) and their normal expression pattern during maturation (n.i. 21 div versus n.i. 5 div). The y-axis reports the log2FC. The x-axis reports DEGs by shL1-a (top) or shL1-b (bottom). **f** Word cloud analysis of the top 50 significant GO terms under the biological process category for up-regulated (top) and down-regulated (bottom) genes. **g** Expression of cell type specific genes in cortical cells expressing shCtrl, shL1-a and shL1-b. RNA levels are normalized on shCtrl. **h** Immunofluorescence staining showing Synapsin-Tomato, Gfap, Gad1 + 2 and NeuroD2 expression. Scale bar = 100 μm. **i** Quantification of cells expressing Gfap, Gad1 + 2 and NeuroD2. For (**c**, **g**, and **i**), $n = 6$ independent samples. Data are mean ± s.e.m. One-way ANOVA with Tukey's multiple comparisons test. Source data are provided as a Source Data file.

*Kdm2a, Kdm2b, Kdm4a, Kdm4b, Kdm5b, Kdm6b, Kmt2b, Kmt2e, Kmt5a, Kmt5b*) and subunits of chromatin remodelling complexes (*Arid1a, Ash2l, Smarcc1, Smarcc2, Smarce1*) including PRC1 (*Bmi1, Pcgf2, Pcgf3*) and PRC2 (*Ezh2* and *Suz12*). GO analysis of these DEGs highlights an enrichment of RNA processing, splicing, transcription, chromatin remodelling and DNA binding (Supplementary Data 3, Fig. 2f, top) terms.

Several mRNAs encoding synaptic proteins were down-regulated by shL1s including *Syn2, Sv2a, Sv2b, Snap25, Syt1, Nrgn, Nrxn1, Cbln4, Syngap1, Lrfn2, Rab3b* and *Camk2* isoforms. GO analysis revealed enrichment in terms related to synaptic transmission, ion transport, channel activity and neurotransmitter receptor activity (Supplementary Data 3, Fig. 2f, bottom). These findings are consistent

with L1 RNAs expression regulating the differentiation of cultured cells by remodelling neuronal transcriptional and chromatin landscapes.

We then investigated if L1 silencing can change the cell type composition of cortical cultures, using the MuSiC deconvolution method. Single-cell drop-sequencing data from mouse postnatal stage 0 (P0) brain cells (Fig. S3D) was used as a reference to estimate differences in cell type proportions in bulk data. MuSiC analysis detected putative markers whose expression changed with shL1-a (Fig. S3E–H), corresponding to three cell clusters with decreased proportions, astrocytes (immature) 2 [13-P] ($P = 1.1e-07$), interneurons 2 [14-P] ($P = 1.78e-06$), and layer V-VI [18-P] ($P = 1.08e-07$), and one cluster with an increased proportion, layer II-IV [4 P] ($P = 0.001$) (Fig. S3I–J). Genes and cell type changes predicted by MuSiC were confirmed by RT-qPCR and immunostaining (Fig. 2g–i). The astrocyte-expressing transcripts for *Slc1a3*, *Aldoc* and *Fabp7* were significantly decreased in shL1-treated samples, as was the number of cells positive for the astrocyte marker protein Gfap, together indicating fewer astrocytes. Expression of *Gad2*, *Sstr3* and serotonin receptors 1 A, 2 A and 2 C mRNAs (Supplementary Data 3) was decreased, and there were fewer cells expressing Gad1+Gad2 proteins, thus indicating a decrease in GABAergic interneurons. Conversely, the increased mRNA expression of *NeuroD2*, *NeuroD6*, *Satb2* and *Pou3f1*, and a greater number of cells positive for NeuroD2 protein revealed a higher number of upper-layer late-born neurons.

To assess the contribution to gene expression changes of reduced L1 retrotransposition activity due to L1 silencing, we treated E17.5 cortical cells with the reverse transcriptase (RT) inhibitors AZT and 3TC from div 5 to div 21 (Fig. S4A). While treatment with 3TC did not significantly alter cells' gene expression profile as measured by RNA-Seq, AZT dysregulated 304 genes (294 down-regulated and 10 up-regulated, FDR < = 0.05, limma voom test; Supplementary Data 5). There was no significant overlap when comparing the 304 genes to those from the shL1-a condition (75 common down-regulated genes and 3 common up-regulated genes, Jaccard index = 0, $P = 0.98$ and $P = 0.65$ respectively, Fisher's exact test) nor to shL1-b ($n = 31$ common down-regulated genes and 5 common up-regulated genes, Jaccard index = 0, $P = 1$ and $P = 0.31$ respectively, Fisher's exact test). There were only 15 down-regulated genes (*C1ql1*, *6030443J06Rik*, *Pcdh15*, *Qk*, *Epn2*, *Zfp521*, *Xylt1*, *Gpm6b*, *Spon1*, *Pdgfra*, *Zcchc24*, *Lhfpl3*, *Sox6*, *Rgcc*, *Asrgl1*) and 2 up-regulated genes (*Lgi2* and *Npas4*) that were consistently affected by the two L1 RNAs silencing and AZT (indicated as consistent at the top of Supplementary Data 5, Fig. S4B–C). These results suggest that L1-RT activity has a limited contribution upon the overall effect on gene expression promoted by L1 RNAs.

The comparison of the effects of L1 RNAs silencing in the two model systems (in vivo and in vitro), allows some inferences about L1 function at different times of corticogenesis. A substantial quantity of genes (367) showed an opposite pattern: they were upregulated in vivo and downregulated in vitro ($P = 9.43e-110$). Their GO terms span from synaptic transmission and signalling to ion transport and neural activity (Supplementary Data 6, Fig. S5A). At the same time, GO analysis of 538 commonly upregulated genes in both systems ($P = 2.2e-16$) revealed a common effect on chromatin remodelling and transcriptional control (Supplementary Data 6, Fig. S5B), including the increased expression of several members of the NeuroD family of pro-neurogenic transcription factors. The impairment of glial cell differentiation in cortical cultures could mimic the effects of depleting developing cerebral cortex of the radial glia, which causes a loss of gliogenic processes after neurogenesis ends.

In summary, L1 RNAs are continuously required for proper corticogenesis, but with different effects on a subset of genes, according to the timing of differentiation of neural progenitors and maturation of single cell types.

## L1 RNAs are mainly associated to chromatin and influence the pattern of H3K27me3 deposition

To further investigate the effects of L1 silencing on transcriptional and chromatin landscapes, we interrogated publicly available ChIP-Seq data from Cistrome DB to infer the transcription factors potentially involved in the regulation of DEGs in E14.5 cortices in vivo, and in cortical cells in vitro (Fig. S5C–D, respectively). Transcription factor binding sites (TFBSs) in DEGs in E14.5 cortex were enriched for the pluripotency markers Myc and Sox2, for the lysine methyltransferase and demethylase Kmt2b and Kdm2b, for Nelf-α and -β, for the neurogenic transcription factors Otx2, NeuroD2, Ascl1, Zic1/2, Smarca4 and for the Polycomb Repressive Complex 2 (PRC2) core components Ezh2 and Suz12 (Fig. 3a). DEGs in cultured neurons showed enrichment of TFBSs for NeuroD2, Smarca4, the core PRC2 components Suz12, Ezh2 and the accessory subunit Jarid2 (Fig. 3b). These results suggest that PRC2 could be involved in the regulation of genes whose expression is influenced in both in vivo and in vitro model systems.

L1 RNAs silencing may cause phenotypic changes through its action on chromatin remodelling complexes and transcription factor activities at multiple levels. The increased expression of the PRC2 core components *Ezh2* and *Suz12* and their neurogenic targets *NeuroD1* and *NeuroD2* (Fig. S5E–F) suggest that one effect of L1 downregulation may be altered expression or activity of epigenetic and transcriptional regulators during neural differentiation, leading to a cascade of expression changes in target genes. L1 might thus act as regulatory RNAs through RNA/protein interactions, like lncRNAs, which bind to chromatin remodelling complexes and transcription factors modulating their activity[18,19].

To test this model, we assessed the localization of L1 transcripts by fractionating cytosolic, nucleoplasmic and chromatinic RNA in cultured cortical cells. The purity of the subcellular fractions was monitored by measuring the enrichment of compartment-specific control RNAs (Fig. S6A). The vast majority of L1 transcripts in 21 div cultures were chromatin-associated ($90.64 \pm 0.63\%$, $92.12 \pm 0.38\%$ and $94.35 \pm 0.32\%$, for L1MdA, L1MdG and L1MdTf respectively), with only a residual percentage of L1 transcripts localized in nucleoplasmic and cytosolic fractions (Fig. 3c). These results are consistent with L1 RNA's association with chromatin in mESCs[9] and with the general localization of COT-I repeat RNA, which includes L1 RNAs[20]. To test the ability of shL1s to reduce the amount of chromatin bound L1 transcripts, we analysed the knockdown efficiency of shL1-a and -b in subcellular RNA fractions. Cytosolic and nucleoplasmic L1 RNAs were both decreased of 60–70%, while chromatinic L1 transcripts by 30–40% (Fig. 3d).

During neuronal development, tri-methylation of histone H3 (H3K27me3) leads to genome-wide transcriptional repression of genes that regulate cell fate transitions and neuronal arborization. Given the impact of L1 RNAs silencing on in vitro neuronal differentiation and their association to chromatin, we investigated by ChIP-seq experiments whether L1 RNAs expression is required, directly or indirectly, for the deposition of the epigenetic mark H3K27me3. L1 RNAs silencing in cultured cortical cells was responsible for an overall higher deposition of H3K27me3 on TSS of down-regulated genes (Fig. 3e), with 4641 genomic regions significantly hypermethylated compared to control cells (FDR < 0.05, DiffBind test; Supplementary Data 7, Fig. 3f). 405 regions out of 4641 were significantly enriched with down-regulated genes ($P = 0.0009$, enrichPeakOverlap test; Supplementary Data 3), while the overlap with up-regulated genes was not significant (139 regions, $P = 0.99$, enrichPeakOverlap). GO analysis of the fraction of down-regulated genes contained in hypermethylated regions revealed an enrichment for biological processes related to neuronal cells' functions and activities including calcium ion concentration, action potential, cell junctions and membrane rafts assembly, neuron projection development, synaptic transmission, assembly, plasticity and potentiation (Supplementary Data 7, Fig. 3g).

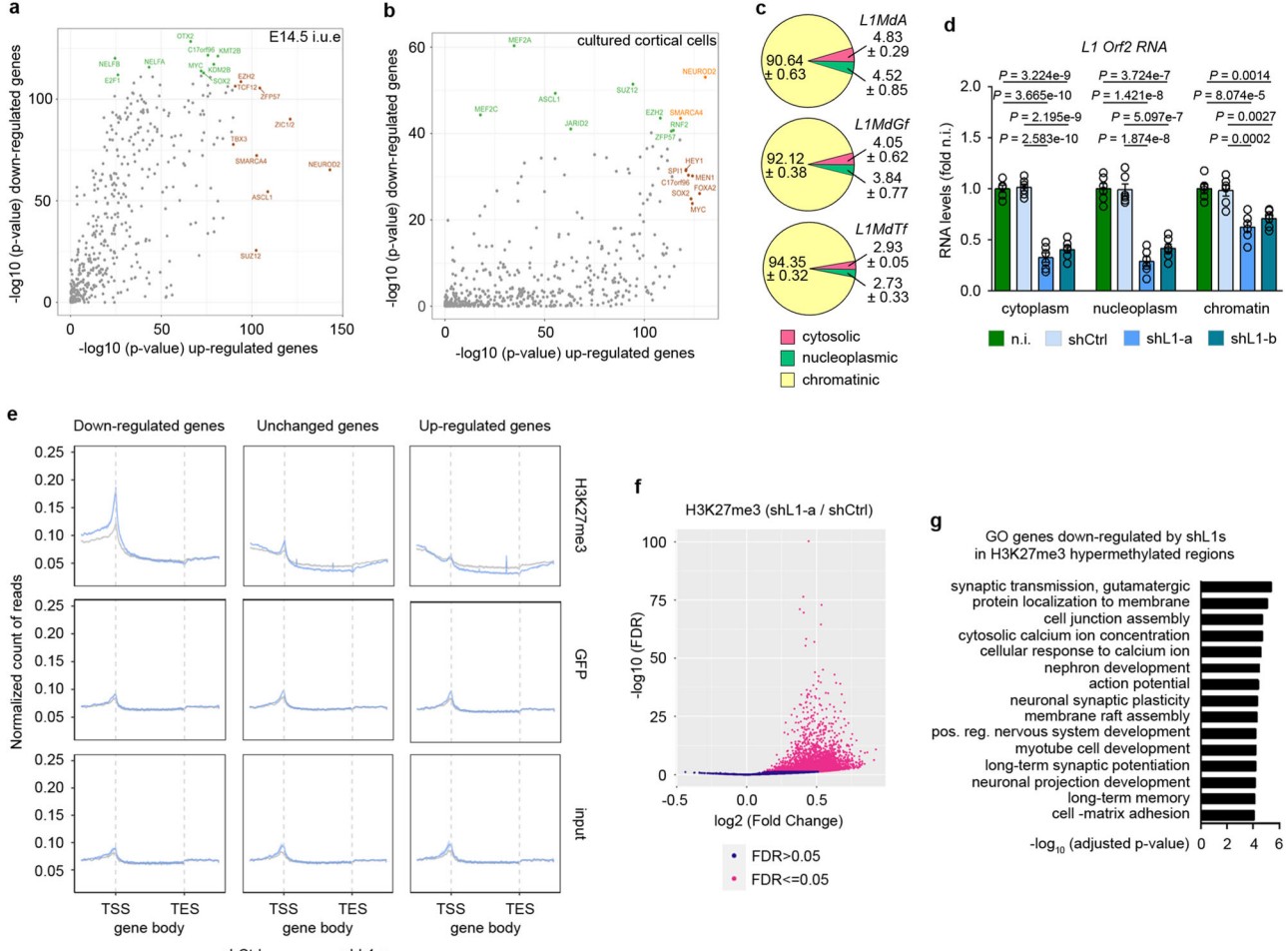

**Fig. 3 | Knockdown of chromatin-associated L1 RNAs influences the signature of H3K27me3 histone mark. a, b** Inference of transcriptional/chromatin regulators in E14.5 mouse cortex electroporated with shL1-a (**a**) and cortical cells infected with shL1-a/b (**b**). The top regulators are in green for the down-regulated gene set and in brown for the up-regulated gene set. Top regulators in common between gene sets are in orange. *p* values were determined by gprofiler2 using a default hypergeometric test and correction for multiple testing has been performed by the g:SCS algorithm. **c** Cytosolic, nucleoplasmic and chromatinic abundance of L1 transcripts in 21 div cells. RNA levels are expressed as percentage of total RNA. Data are mean ± s.e.m. *n* = 3 independent biological replicates. **d** Expression of L1 transcripts in cytoplasmic, nucleoplasmic and chromatinic fractions of 21 div cells non infected (n.i.) or infected with shCtrl, shL1-a and shL1-b. RNA levels are normalized on the n.i. group for each fraction. Data are mean ± s.e.m. *n* = 6 independent biological samples. One-way ANOVA with Tukey's multiple

comparison test. **e** Metagene plot showing H3K27me3, GFP and input ChIP-seq signals for down-regulated, unchanged and up-regulated genes in mouse cortical cells infected with shCtrl or shL1-a. The plot shows the average signals of *n* = 2 independent biological replicates for each condition with 95% intervals of confidence as shaded areas. **f** Volcano plots representing differentially enriched ChIP-seq peaks in shL1-a cortical cells for H3K27me3. X-axis shows the log2 (Fold Change). The *Y*-axis shows the -log₁₀ (FDR). *n* = 2 independent biological replicates. **g** Top GO terms under the biological process category for genes down-regulated by shL1s according to RNA-seq and contained in peaks/islands with a significantly higher deposition of H3K27me3 after shL1-a. *p* values were determined by gprofiler2 using a default hypergeometric test and correction for multiple testing has been performed by the g:SCS algorithm. Source data are provided as a Source Data file.

These results suggest that changes in repressive epigenetic marks may account for a portion of the effects on gene expression observed after L1 RNAs silencing in cultured cortical cells.

## L1 transcripts are bound by PRC2 proteins Suz12 and Ezh2

We then searched for putative protein interactors of L1 RNAs to determine whether transcription factors and chromatin remodelers involved in L1-dependent expression patterns can bind L1 RNAs. 508 full-length L1 mouse genomic sequences representing the main L1 subfamilies (L1Lx, L1MdV, L1MdFanc, L1MdMus, L1MdF, L1MdA, L1MdGf and L1MdTf) (Supplementary Data 8 and Fig. S6B–C) were submitted to the *cat*RAPID omics *V2* algorithm[21]. Combining secondary structure, hydrogen bonding, van der Waals contributions and experimental enhanced crosslinking and immunoprecipitation (eCLIP) data, *cat*RAPID estimates the binding propensity of protein-RNA pairs. The PRC2 component Suz12 showed the highest interaction

propensity of all the 1946 RNA binding proteins (RBPs) in the *cat*RAPID database (Supplementary Data 9). While Suz12 binding motifs were found in all L1 subfamilies, the most recently evolved L1MdGf and L1MdTf showed a higher percentage of single L1 sequences with Suz12 binding sites (100% of the sequences tested, Fig. 4a) and the highest number of Suz12 binding motifs per sequence (Fig. 4b). Another PRC2 subunit, Ezh2, was also a high-score putative interactor, with an increasing propensity in the most evolutionarily recent L1 subfamilies (Fig. 4c).

PRC2 is a major regulator of the balance between self-renewal and differentiation of multipotent progenitor cells[22,23]. PRC2 inhibits the expression of many genes involved in neuronal development[24] acting as a histone methyltransferase that trimethylates K27 of histone H3 (H3K27me3)[25]. Ezh2 is the catalytic enzyme, whereas Suz12 and Eed act as essential regulatory subunits. PRC2 binding to chromatin requires RNA[26], interacting preferably with RNA G-quadruplex secondary

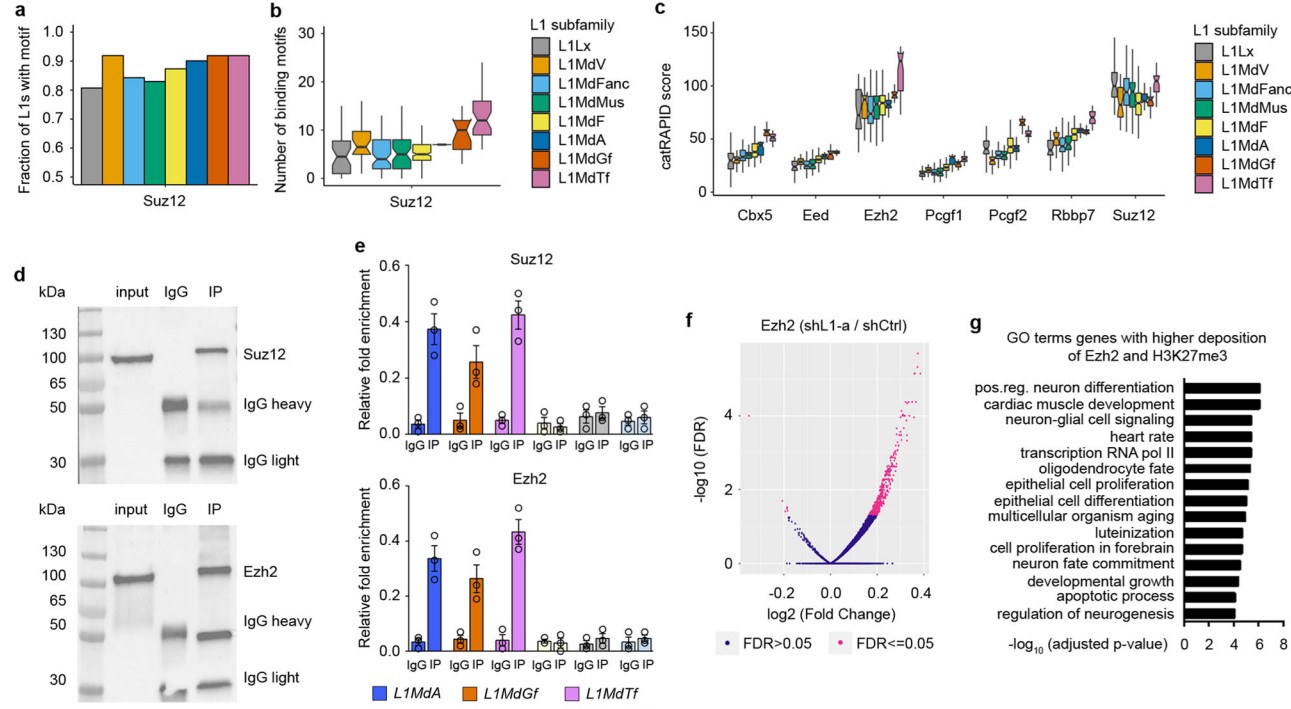

**Fig. 4 | L1 RNAs bind PRC2 subunits Ezh2 and Suz12 and influences the binding of Ezh2. a, b** Boxplots showing the fraction of L1 sequences per evolutionary time-point hosting a Suz12 RNA binding motif (**a**) and the number of motifs per L1 detected by *cat*RAPID (**b**). **c** Boxplots of *cat*RAPID score for 7 Polycomb proteins along evolutionary time-points. The boxes show the interquartile range (IQR), the central line represents the median, the whiskers add 1.5 times the IQR to the 75 percentile (box upper limit) and subtract 1.5 times the IQR from the 25 percentile (box lower limit). For (**a**, **b** and **c**), the number of different L1 elements for each time-point is the following: L1Lx $n = 78$, L1MdV $n = 26$, L1MdFanc $n = 38$, L1MdMus $n = 17$, L1MdF $n = 205$, L1MdA $n = 47$, L1MdGf $n = 47$, L1MdTf $n = 57$. **d, e** RNA immunoprecipitation (IP) for Suz12 and Ezh2 with chromatinic RNA fraction of 21 div cells. Western blot showing Suz12 and Ezh2 enrichment after IP (**d**) and RNA levels by RT-qPCR for L1 subfamilies RNAs, *Gapdh*, *45 s rRNA* and *U1 snRNA* in Suz12 or Ezh2 IP. RNA levels are relative to input control. $n = 3$ independent biological samples. Data are mean ± s.e.m. **f** Volcano plots representing differentially enriched ChIP-seq peaks in shL1-a cortical cells for Ezh2. X-axis shows the log2 (Fold Change). The *Y*-axis shows the -log$_{10}$ (FDR). $n = 2$ independent biological replicates. **g** Top GO terms under the biological process category for genes contained in peaks/islands characterized by a significantly higher deposition of Ezh2 and H3K27me3. *p* values were determined by gprofiler2 using a default hypergeometric test and correction for multiple testing has been performed by the g:SCS algorithm. Source data are provided as a Source Data file.

structures (G4s) and G-tracts. The high abundance of these motifs contributes to the promiscuity of PRC2-RNA interactions[27,28]. In turn, RNA can modulate the enzymatic activity of chromatin-bound PRC2[29].

We thus validated the binding of Suz12 and Ezh2 to L1 RNAs in 21 div cortical cells by native RNA immunoprecipitation (RIP). L1MdA, L1MdGf and L1MdTf transcripts were detected by RT-qPCR both in Suz12 and Ezh2 immunoprecipitate of whole cell lysate or chromatinic fraction while negative controls *Gapdh*, *45s RNA* and *U1 snRNA* were not enriched (Fig. S6D−E and Fig. 4d, e). RIP with an antibody against the known L1 RNA interactor Nucleolin for L1MdA, L1MdGf, L1MdT transcripts and *45s rRNA* served as a positive control (Fig. S6F)[9].

Ezh2 binds G-rich tracts in cellular transcripts that are prone to forming G4s[30,31], leading to displacement of PRC2 from nucleosomes and to H3K27me3 depletion from genes, thus regulating their expression[27]. To test whether G4s could be involved in the binding of Suz12 and Ezh2 to L1 RNAs, we overlapped the binding sites found by *cat*RAPID and the predicted G4 sequences in L1 subfamilies. Putative Suz12 binding sites are spread throughout L1s and a significant overlap with G4s was observed only for L1MdFanc ($P = 0.0089$), L1MdF ($P = 5.3e−13$), L1MdMus ($P = 0.0028$) and L1MdTf ($P = 3.3e−12$), while no significant correlation was found for other L1 families (Fig. S6G)[32]. In contrast, the putative binding sites for Ezh2 are in the 3'UTR of all the L1 subfamilies tested, which is also the region highly enriched for G4 prone sequences[33]. In fact, the overlap between the binding sites for Ezh2 and the predicted G4s is highly significant (Fig. S6H). Interestingly, binding sites for Ezh2 and Suz12 did not coincide, confirming cross-linking data on total RNA[34].

These findings suggest that binding to L1 transcripts could be important for regulating PRC2 activities.

## L1 silencing influences Ezh2 activity to target genes in cortical cultured neurons

We then investigated by ChIP-seq experiments whether L1 RNAs expression is required for PRC2 binding to target genes and for the consequent deposition of the epigenetic H3K27me3 mark.

L1 RNAs silencing in cultured cortical cells was responsible for a higher binding of Ezh2 in 412 regions (FDR < 0.05, DiffBind test; Supplementary Data 7, Fig. 4f). The majority of Ezh2 bound regions, 261 (62%; $P = 7.6E−11$), overlapped with H3K27me3 hypermethylated sites, suggesting that L1 RNAs knockdown was able to alter, at least for these sequences, both Ezh2 occupancy and H3K27me3 deposition. Interestingly, genes affected by both increased deposition of Ezh2 and H3K27me3 mark were important for transcriptional regulation of neuronal fate commitment and glial and oligodendrocyte lineages specification (*Sox1*, *Sox2*, *Sox3*, *Sox6*, *Sox7*, *Pax3*, *Otx1*, *Otx2*, *Hes1*, *Tbr2*, *Nkx1-2*, *Nkx2-2*, *Nkx2-9*, *Stat3*, *Stat5a*, *Stat5b*, *Olig3*, *Wnt5a*, *Wnt7a*, *Bmp4*, *Bmp6*) as well as for neuronal cell functionalities, including development of neuronal projections and synaptic processes (*Grid2*, *Bdnf*, *Gad1*, *Slc10a4*, *Slc26a4*, *Slc6a4*, *Slc16a2*, *Pmp22*, *Camkmt*, *Adra1a*, *Adra1b*) (Supplementary Data 7, Fig. 4g). 44 of the regions with an overall higher binding of Ezh2 were significantly enriched with down-regulated genes ($P = 0.03$, enrichPeakOverlap test), while no significant overlap was observed with up-regulated genes ($P = 0.75$, enrichPeakOverlap test).

These results suggest that in in vitro cultured mouse cortical cells L1 RNAs may influence PRC2 activity by regulating the binding of Ezh2 and by affecting the consequent deposition of the repressive histone modification H3K27me3 to regulatory regions of a group of genes whose expression is crucial for controlling neuronal and glial cell commitment and differentiation.

## Discussion

Our study finds that L1s have a role as regulatory lncRNAs in corticogenesis. Like for ESCs, L1 RNAs expression in the developing mouse brain cortex is required for a proper balance between proliferation and differentiation in neuronal progenitors, contributing to cell maintenance in a growing state[9]. In this context, L1 RNAs promote the synthesis of an efficient translational apparatus and of nuclear-encoded mitochondrial proteins to sustain cellular growth.

L1 RNAs are also required to establish the proper transcriptional cascade leading to neuronal differentiation and maturation, for a correct repertoire of neuronal cell types and for newborn neurons to correctly migrate along the cortical plate. Accordingly, L1 silencing affects corticogenesis by remodelling neuronal transcriptional and chromatin landscapes depending on the timing of differentiation and by compromising cell maturation.

It remains to be determined how L1s can promote both neuronal progenitor proliferation in vivo and neuronal differentiation in vitro. This is mirrored by the opposite pattern of expression of genes that were upregulated in vivo and downregulated in vitro and involved in synaptic transmission, signalling, ion transport and neural activity (Supplementary Data 6, Fig. S5A). This behaviour could be the consequence of the time of harvesting of cortical neurons for in vitro differentiation (E17.5), modelling later stages of the neurodevelopmental cascade with respect to the E12.5 stage tested in vivo. Furthermore, the time required for proper AAV-mediated expression of shL1s in cultured cells may delay further the consequences on neuronal maturation and cell type composition. Most of the pro-neurogenic effect in vivo could be due to the impact of L1 silencing upon proliferation and commitment of progenitor cells, which are much less represented at E17.5. Importantly, the difference in the effect of L1 silencing may be caused by the heterogeneity of L1 transcripts. Their biological activity may depend on the location in the genome (i. e. intergenic, intronic, promoter or enhancer-associated), length (full length, 5' and 3' truncated), protein interaction network and expression, being differentially regulated according to the TFBS content of their promoters. By taking advantage of targeting a large repertory of heterogeneous L1 RNAs, the complexity of the regulated L1 transcriptome and the function associated to single transcripts, are missed. Unveiling the repertory of single L1 transcripts by third generation sequencing technologies, will be instrumental to better define their specific functions in the two model systems.

L1s may exert regulatory activities through several mechanisms. While the role of L1s mobilization is unlikely given the lack of substantial consequences of RT inhibition, the synthesis of ORF2p could enhance transcription of pro-neural genes through a topoisomerase-like effect on the chromatin structure by its endonuclease activity[12]. Nevertheless, the outcomes of L1 RNAs silencing seem to be mainly associated to activities as regulatory lncRNAs and in particular to their binding to chromatin. According to a model of action, L1 RNAs may cause phenotypic changes through their direct or indirect impacts on chromatin remodelling complexes and transcription factors activities, as found for lncRNAs[18]. Binding sites for pro-neurogenic transcription factors were indeed enriched in regulatory regions of genes differentially expressed in vivo and in vitro upon L1 RNAs silencing. A large number of TSS of genes down-regulated according to RNA-seq experiments, were hypermethylated at H3K27me3, a repressive epigenetic mark. To understand how L1 RNAs may influence these nuclear activities, protein interactors were predicted computationally and validated in silico for the presence of experimental eCLIP data. They provide candidate regulatory networks to be further studied for the functional outputs of L1 RNAs expression during cortical development.

Different lines of evidence suggest a portion of them may involve PRC2. DEGs upon L1 RNAs downregulation are enriched for binding sites for PRC2 components both in vivo and in vitro. Computational predictions and direct experimental validation show that chromatin-associated L1 RNAs can bind both Ezh2 and Suz12, providing a potential mechanism for the influence of L1 RNAs on PRC2 activity. Importantly, L1 RNAs silencing influences Ezh2 binding and H3K27me3 deposition at a pool of genes involved in neuronal differentiation.

In ESCs, PRC2 chromatin binding requires RNA: a perturbation of this interaction results in a substantial loss of PRC2 occupancy at its canonical targets, and a gain at some de novo targets[26]. RNA may have multiple roles for PRC2 activity: PRC2 recruitment during gene inactivation and eviction during gene reactivation as well as regulation of its enzymatic activity[29]. PRC2 interacts preferably with RNA G4s and G-tracts and the high abundance of these motifs contributes to the promiscuity of PRC2-RNA interactions[31]. G4 RNAs may evict PRC2 from chromatin during gene activation, providing a 'sponge' to draw PRC2 away from repressive interactions.

Our computational predictions of L1 RNAs binding sites for Suz12 and Ezh2 recapitulate the main experimental evidence for their interaction with cellular RNAs. For Suz12, sites are widely distributed along L1 sequences with no overlap with G4s or Ezh2 binding sites. On the contrary, L1s have a restricted accumulation of Ezh2 binding sites at the 3' end overlapping G4 sequences. These data suggest that L1 RNAs may influence PRC2-dependent deposition of repressive epigenetic marks on genes crucial for neuronal cell differentiation and activity, a prediction that has been validated with ChIP-seq experiments. Future work will address the details and functional outputs of PRC2-L1 RNAs interactions at single gene loci and the significance of the difference in the magnitude of variations between H3K27me3 deposition and Ezh2 binding upon L1 RNAs silencing.

In summary, temporal-specific expression of transposable elements from the non-coding genome may contain the instructions to control epigenetic regulation and stage-specific progression of a cell through the steps of corticogenesis. Our work highlights another evolutionary innovation which might have emerged from the interaction between transposons and the host genome through the tree of life. L1s might have evolved to act as PRC2 decoy to allow cells to escape differentiation and maintain them in a proliferative status, in turn increasing chances for retrotransposition. The genome might have exploited this selfish behavior to increase complexity in the brain, allowing the addition of layers of differentiated cells while ensuring the maintenance of a pool of proliferative neural precursors. All this could have been achieved by finely modulating L1 expression and their influence on PRC2-dependent gene networks given the distinct distribution of predicted Suz12 and Ezh2 binding sites in L1 elements of different evolutionary ages. These results suggest a fascinating scenario in which L1s and PRC2 have co-evolved to regulate developmental processes in the brain.

Since G4s operate as common binding hubs for many transcription factors to promote increased transcription of G4-containing genes[32–34], we speculate that L1 RNAs can influence the activity of other transcriptional networks by similar mechanisms and that these interactions have been relevant in the evolution of the brain.

## Methods
### Animals
C57BL/6 J and CD1 IGS mice (Charles River) were housed at *Istituto Italiano di Tecnologia* (IIT). All animal procedures were approved by IIT animal use committee and the Italian Ministry of Health (Animal Study Proposal #693/2019-PR.) and conducted in accordance with the Guide for the Care and Use of Laboratory Animals of the European

Community Council Directives. All mice were group-housed under a 12-hours light-dark cycle in a temperature and humidity-controlled environment (18–23 °C and 40–60%, respectively) with ad libitum access to food and water.

## Plasmids, cloning and adeno-associated vectors (AAV)

Short hairpin sequences targeting a conserved region of L1-Orf2 were cloned in a pLKO-RFP-shCtrl plasmid (Addgene, #69040) using Asp718I and EcoRI restriction sites to test the knockdown efficiency in mouse cultured cells lines (data not shown). For in utero electroporation experiments, control shRNA (shCtrl), L1 shRNA-a (shL1-a) and L1 shRNA-b (shL1-b) were cloned along the hU6 promoter in a pSilencer 3.0-H1 (ThermoFisher Scientific) plasmid using EcoRI and HindIII restriction sites and substituting the original H1 promoter. To visualize electroporated cells, pSilencer 3.0-U6-shCTRL, pSilencer 3.0-U6-shL1-a and pSilencer 3.0-U6-shL1-b vectors were co-electroporated together with a pCAGGS-IRES-GFP vector with a 1:1 molar ratio. For AAV generation, the U6 shRNA expression cassette was PCR amplified from pRNAT-U6 plasmid (Genescript) and cloned into pAAV-hSyn-TdTomato vector (an AAV vector plasmid derived from AAV-hSyn-EGFP, Addgene #50465). The shRNA sequences (shCtrl, shL1-a, or shL1-b) were cloned downstream of the U6 promoter using BamHI and HindIII restriction sites. AAV particles were produced as in referenced in[35]. All plasmids used were successfully verified by Sanger sequencing.

## In utero electroporation

Animal care and experimental procedures were performed in accordance with the IIT licensing and the Italian Ministry of Health license n° 176AA.68. E12.5 timed-pregnant CD1 IGS mice were anesthetized with isoflurane and administered 2–5 mg/kg ketoprofene for analgesia. The uterine horns were exposed by laparotomy and the DNA (1 µg/µl) with 0.01% Fast Green dye (Sigma Aldrich) was injected in the lateral ventricle using a glass capillary (B100-58-10, Sutter Instrument). 3 mm diameter platinum tweezer electrodes (CUY650P3, NepaGene) were used to electroporate the cerebral cortex. Four electrical pulses (33 V, 30 msec duration, 970 msec interval) were delivered using a NEPA21 electroporator (NEPA21, NepaGene).

## Immunofluorescence and imaging

Mice were sacrificed at indicated ages and the brains dissected and fixed in 4% paraformaldehyde (PFA) (Sigma Aldrich) at 4 °C overnight. Brains were de-hydrated in 30% sucrose and sliced in 20 µm coronal cryosections using a CM 3050 S cryostat (Leica). Slices were permeabilized and blocked in 1x PBS containing 5% normal goat serum (NGS, Abcam) and 0.3% Triton X-100 (Sigma Aldrich). Sections were afterwards incubated with primary antibodies (see Supplementary Data 1) diluted in blocking solution overnight at 4 °C. After an extensive wash in 1x PBS + 0.1% Triton X-100 (PBST) brain slices were incubated with fluorescent dye conjugated secondary antibodies (see Supplementary Data 1) diluted in blocking solution for 1 h at RT. Slices were counterstained with Hoechst 33342 1:10.000 in PBST (Thermo Fisher Scientific) for 20 min and extensively washed in 1x PBS, mounted with Mowiol (Sigma Aldrich) and examined with confocal microscopy. Fluorescent images were acquired with Nikon A1 confocal microscope equipped with a 20x objective and analyzed with Nikon software version 4.11.0 (NIS Elements). Positive cells for the indicated marker were counted through the depth of the cortex in the electroporated area and the percentages normalized on the total number of GFP⁺ cells as indicated in the figure legends. For cell number quantifications, all relevant sections containing GFP⁺ electroporated cells from rostral to caudal were quantified upon shCtrl and shL1-a conditions by ImageJ version 1.53i (Wayne Rasband, National Institutes of Health, USA). In order to assess GFP⁺ cells distribution at E18.5, the neocortex was divided radially in 10 equal-sized bins from the pia to the upper edge of the white matter. The cells in each bin were quantified and reported as the percentage of total counted cells.

## FAC-sorting of GFP⁺ cells from i.u.e. mouse brain cortex

Pregnant female mice electroporated at E12.5 with shCtrl or shL1-a, were sacrificed at E14.5 and brain cortices (n = 2 for both shCtrl and shL1-a) dissected under a stereomicroscope in ice-cold HBSS and enzymatically dissociated using a Neural Tissue Dissociation Kit (Miltenyi Biotec) following manufacturer's protocol. After tissue digestion at 37 °C, cells were manually dissociated by pipetting, filtered with a 40 µm cell strainer, centrifuged for 10 min at 300 x g and resuspended in ice-cold HBSS. GFP⁺ cells were FAC-sorted with a SH800S instrument (Sony). For each shCtrl and shL1-a samples, n = 3 pools of 200 GFP⁺ cells were collected for direct RNA-Seq library preparation using the SMART-Seq HT PLUS kit (Takara), according to manufacturer's instructions

## Neuronal cultures, AAV infection and treatment with reverse transcriptase inhibitors

Dissociated cortical neurons from E17.5 C57BL/6 J embryos were plated at a concentration of 50,000 cells/cm² onto poly-D-lysine (0.5 mg/ml, 1 h at 37 °C) coated dishes and maintained in Neurobasal medium supplemented with 1% glutamax, 1% penicillin/streptomycin and 2% B27 (all by Thermo Fisher Scientific), in a humified atmosphere with 5% CO₂ at 37 °C for 21 days (div; days in vitro). Fifty percent of the medium was changed every 6 days. Cells were collected and tested at 3, 7, 14 and 21 div. 5 div aged neuronal cells were infected for 8 hours with AAV-hSyn-tdTomato-U6-shCtrl, AAV-hSyn-tdTomato-U6-shL1-a or AAV-hSyn-tdTomato-U6-shL1-b with a M.O.I. (multiplicity of infection) of 100. After the infection, culture medium containing viral particles was replaced with fresh medium and cells were grown until 21 div. Inhibition of L1 reverse transcriptase was performed by treating 5 div aged cultured cells with 2 µM AZT, 5 µM 3TC, or water as a vehicle, until 21 div. Half of the cell culture medium was replaced with fresh new medium containing the drug every three days from the beginning of the treatment.

## RNA purification and RT-qPCR

Total RNA from mouse cortex, or in vitro cultured cortical cells, was purified by TRIzol Reagent (Thermo Fisher Scientific) and DNA was removed by treatment with DNAse I (Sigma-Aldrich). cDNA was synthesized from 0.5 µg of RNA using an iScript cDNA Synthesis kit (Bio-Rad). Real time quantitative PCR was performed on a CFX96 Touch™ Real-Time PCR Detection System (Bio-Rad) with CFX Maestro Software 2.3. For L1 detection, duplex TaqMan reactions were performed with Ubiquitin C (UbC), Ywhaz, or TATA-binding protein (Tbp), as reference genes using an iQ Multiplex Powermix (Bio-Rad). For all the other targets, a SsoAdvanced Universal SYBR Green Supermix (Bio-Rad) was used with UbC, Tbp, Ywhaz or Gapdh as reference genes. For both TaqMan and SYBR Green reactions, cycling conditions were: 95 °C for 20 s, followed 40 cycles of 95 °C for 10 s and 60 °C for 1 min. No template and no RT controls were included. Expression levels were determined relative to the reference gene using the ΔΔCt method.

## Bisulfite sequencing

Direct bisulfite sequencing was performed as previously described[36]. DNA from cultured cortical cells was purified by DNeasy Blood and Tissue kit (Qiagen) and bisulfite conversion performed using an Epitect Bisulfite kit (Qiagen). L1MdTf promoter was amplified by PCR using ZymoTaq pre-mix (Zymo Research) with the following cycling conditions: 95 °C 10 min, 50 cycles of 94 °C for 30 s, 60 °C for 40 s and 72 °C for 1 min, followed by 72 °C hold for 7 min. PCR products were cleaned up with ExoSAP-IT Reagent (Thermo Fisher Scientific) and sequenced with an ABI PRISM 3100 Genetic Analyzer (Thermo Fisher Scientific) using the reverse primer. Peak heights from resulting chromatograms

were analyzed using Sequencing Analysis Software for Windows 10. Percent methylation was measured by comparing peak heights for G and A (reverse strand sequencing). Standard of known methylation level (EpiGentek) were treated and analyzed identically to test samples to confirm complete bisulfite conversion.

## Chromatin immunoprecipitation

For chromatin immunoprecipitation, $20 \times 10^6$ cortical cells were fixed with 1% formaldehyde, scraped and washed twice with PBS by centrifugation at 1.600 rpm for 5 min. Cell pellets were resuspended in 10 mL of Nuclear Extraction Buffer 1 (50 mM HEPES-KOH pH 7.5, 140 mM NaCl, 1 mM EDTA, 10% glycerol, 0,5% NP-40, 0.25% Triton X-100), rocked at 4 °C for 10 min and spin at $2.000 \times g$ for 4 min at 4 °C. Cells were then resuspended in 10 mL of Nuclear Extraction Buffer 2 (10 mM Tris-HCl pH 8.0, 200 mM NaCl, 1 mM EDTA, 0.5 mM EGTA), rocked gently at 4 °C for 5 min and spin at $2.000 \times g$ for 5 min at 4 °C. Nuclei were obtained by centrifugation at $2.000 \times g$ for 5 min at 4 °C, and resuspended in 600 μL SDS Lysis Buffer (1% SDS, 10 mM EDTA, 50 mM Tris-HCl pH8.0) supplemented with Complete Protease Inhibitor cocktail (Sigma Aldrich). DNA was sheared with a Bioruptor Pico sonication device (Diagenode) for 30 cycles (30" ON, 30" OFF). For the assessment of chromatin fragmentation, 175 μL of De-Crosslink Buffer (1% SDS, 0.1 M NaHCO₃) were added to 25 μL lysate and boiled at 95 °C for 5 min. DNA was purified with Phenol/Chlorophorm/Isoamyl and run on a 2100 Bioanalyzer Instrument (Agilent). Lysates were immunocleared by incubation for 1 h with 20 μL Protein A + 20 μL Protein G, previously blocked for 1 h with Beads Blocking Buffer (10 mM Tris-HCl pH 7.4, 1 mM EDTA, 0.5 mg/ml BSA). 25 μL of immunocleared lysate were saved as input sample. For the immunoprecipitation, 8 μg of antibody (H3K27me3, Ezh2, GFP, H3K9me3 or IgG) were added to 100 μL of immunocleared lysate and incubated over-night at 4 °C on a rotating wheel. Samples were centrifuged at $13.000 \times g$ for 20 min at 4 °C to remove aggregates and incubated with a mix of 15 μL of Protein A and 15 μL of Protein G (previously blocked with Beads Blocking Buffer, 1 h at 4 °C) for 2 h at 4 on a rotating wheel. Beads-immunocomplexes were washed on a magnetic rack as follow: 2 times with 1 mL of Mixed Micelle Wash Buffer (150 mM NaCl, 20 mM Tris-HCl pH 8.0, 1 mM EDTA, 5.2% w/v sucrose, 0.02% NaN₃, 1% Triton X-100, 0.2%SDS), 2 times with 1 mL Wash Buffer 500 (0.1% w/v deoxycholic acid, 1 mM EDTA, 50 mM HEPES pH 7.5, 500 mM NaCl, 1% Triton X-100, 0.02% NaN₃), 2 times with 1 mL LiCl Detergent Buffer (0.5% w/v deoxycholic acid, 1 mM EDTA, 250 mM LiCl, 0.5% v/v NP-40, 10 mM Tris-HCl pH 7.5, 0.02% NaN₃), once with 1 mL of TE (10 mM Tris-HCl pH 7.4, 1 mM EDTA). Both immunoprecipitates and input samples were de-crosslinked by adding 130 μL of De-Crosslink Buffer over-night at 65 °C. DNA was purified with Zymo ChIP DNA Clean and Concentrator kit (Zymo Research) and eluted in 12 μL of Elution Buffer. DNA quality was evaluated on a 2100 Bioanalyzer Instrument (Agilent). Purified DNA was used for preparation of ChIP-seq libraries or for each qPCR reaction (2 μL per reaction). Enrichment levels were calculated as a percentage of the input.

## Subcellular fractionation and RNA localization

Primary mouse cortical cells cultured for 21 div were fractionated according to a previously published protocol[37] with minor modifications. Briefly, cells were rinsed twice with cold PBS, scraped with 400 μL cell lysis buffer (10 mM Tris pH 7.4, 150 mM NaCl, 0.15% Igepal CA-630), collected in a 1.5 mL eppendorf tube and incubated on ice for 5 min. After the incubation, cell lysates were overlayed on the top of a 1 mL sucrose buffer (10 mM Tris pH 7.4, 150 mM NaCl, 24% sucrose) and centrifuged at $1000 \times g$ for 10 min at 4 °C. Supernatants, corresponding to the cytosolic fraction, were cleared by centrifugation at $14,000 \times g$ for 1 min, and stored on ice. Nuclei pellets were washed twice with ice-cold PBS-EDTA, resuspended in 100 μL glycerol buffer (20 mM Tris pH 7.4, 75 mM NaCl, 0.5 mM EDTA, 50% Glycerol) followed

by 100 μL of nuclear lysis buffer (10 mM Tris pH 7.4, 1 M Urea, 0.3 M NaCl, 7.5 mM MgCl₂, 0.2 mM EDTA, 1% Igepal CA-630) and incubated for 5 min on ice. Lysates were centrifuged at $13,000 \times g$ for 2 min to precipitate chromatin-RNA complexes. Supernatants (corresponding to the nucleoplasmic fraction) were collected and stored on ice. Chromatin pellets were washed twice with ice-cold PBS-EDTA and kept in ice. RNA was purified from cytosolic, nucleoplasmic and chromatinic fractions with TRIzol Reagent (Thermo Fisher Scientific), digested with DNase I (Sigma Aldrich) and reverse transcribed with the iScript cDNA Synthesis kit (Bio-Rad). Samples treated, or un-treated, with the reverse transcriptase (RT+ or RT−, respectively) were both loaded to control possible residual DNA contamination. L1 subfamilies RNA abundance relative to the cytosolic fraction was determined by qPCR. 45 s pre-ribosomal, Gapdh, 7sL and Cytochrome b RNAs were used to assess proper fractionation enrichment.

## Immunofluorescence and imaging for in vitro cultured neurons

Mouse embryonic (E17.5) cortical cells were isolated and cultured onto poly-D-lysine coated coverslips, as previously described. After 21 div, cells were washed with PBS, fixed with 4% PFA for 10 min, permeabilized by three washes of 15 min each with PBS + 0.1% Triton X-100 (PBS-T, Sigma Aldrich), blocked for 1 h at room temperature with 5% NGS (Abcam) in PBS-T and incubated with primary antibodies (see Supplementary Data 1) in blocking solution over-night at 4 °C. Cells were washed three times with PBS-T and then incubated with fluorescent dye conjugated secondary antibodies (see Supplementary Data 1) in blocking solution for 1 h at room temperature. After three washes with PBS-T, cells were counterstained with Hoechst 33342 (1:10.000 in PBST) (Thermo Fisher Scientific) for 20 min and extensively washed in 1x PBS, mounted with Mowiol (Sigma Aldrich) and examined with confocal microscopy. Fluorescent images were acquired with Nikon A1 confocal microscope equipped with a 40x objective and analyzed with Nikon software version 4.11.0 (NIS Elements) and ImageJ version 1.53i (Wayne Rasband, National Institutes of Health, USA).

## RNA immunoprecipitation

To assess protein-RNA interaction, we performed RNA immunoprecipitation (RIP) experiments in native conditions on both the total cell lysate and the chromatinic fraction, according to previously published procedures[38–40]. For IP on the total cell lysate, $10 \times 10^6$ 21 div mouse cortical cells were scraped with 500 μL of polysome lysis buffer (10X stock solution: 1000 mM KCl, 50 Mm MgCl 2, 100 mM HEPES-NaOH pH 7, 5% NP-40), pipetted up and down on ice to promote cell lysis and centrifuged at $20,000 \times g$ for 10 min at 4 °C. For IP on the chromatin fraction, nuclei from $10 \times 10^6$ 21 div mouse cortical cells were purified as in (4) and chromatin released by incubation for 10 min on ice in nuclear extraction buffer 1 (50 mM HEPES-KOH pH 7.5, 140 mM NaCl, 1 mM EDTA, 10% glycerol, 0.5% NP-40 and 0.25% Triton X-100), and for 5 min on ice in nuclear extraction buffer 2 (10 mM Tris-HCl pH 8.0, 200 mM NaCl, 1 mM EDTA, 0.5 mM EGTA), centrifuged at $2.000 \times g$ for 5 min, resuspended in nuclear resuspension buffer (50 mM HEPES-NaOH pH 7, 10 mM MgCl₂) and treated with DNase I (Sigma-Aldrich). 100 μL of supernatant were used for each immunoprecipitation reaction and 10 μL were collected in a new tube as input. Protein A-coupled Dynabeads (Thermo Fisher Scientific) were incubated for 1 h at room temperature with 5 μg of Suz12, Ezh2, Nucleolin, or Rabbit IgG, antibodies (see Supplementary Data 1), washed five times with NT-2 buffer (5X stock solution: 250 mM Tris-HCl pH 7.4, 750 mM NaCl, 5 mM MgCl₂, 0.25% NP-40) and then incubated with 100 μL of cell lysate on a rotating wheel o/n at 4 °C. After the incubation, samples were washed five times with NT-2 buffer, treated with proteinase K at 55 °C for 30 min, and RNA purified with TRIzol Reagent (Thermo Fisher Scientific), digested with DNase I (Sigma Aldrich) and reverse transcribed with iScript cDNA Synthesis kit (Bio-Rad). Enrichment levels relative to input were calculated by qPCR. Samples treated, or un-treated, with

the reverse transcriptase (RT+ or RT−, respectively) were both loaded to control possible residual DNA contamination.

## Immunoblotting

Input, IgG isotype negative control and immunoprecipitated (IP) samples were collected after the RNA immunoprecipitation protocol. Protein samples were obtained by lysis in RIPA buffer (1% Triton X-100, 150 mM NaCl, 20 mM $Na_2PO_4$, pH 7.4) supplemented with Halt Protease and Phosphatase Inhibitor Cocktail (Thermo Fisher Scientific). To determine protein concentration, a BCA assay (Thermo Fisher Scientific) was performed, and protein samples were separated by SDS-PAGE followed by semi-dry transfer to nitrocellulose membranes. Blots were blocked with 5% non-fat milk and then incubated at 4 °C overnight with primary antibodies (see Supplementary Data 1). After three washes, blots were incubated with HRP-linked secondary antibodies (see Supplementary Data 1) and signal detected with SuperSignal West Pico Plus chemiluminescent substrate (Thermo Fisher Scientific) and imaged in a iBright 1500 instrument (Thermo Fisher Scientific) with a iBright Analysis Software.

## Computational characterization of LINE-1 interactions

*cat*RAPID omics *V2*[21,41] was used to characterize the interactions between mouse RNA Binding Proteins (RBPs) and L1 sequences in terms of Interaction propensities and the presence of RNA binding Motifs searched within the RNA targets of the proteins. Among RBPs that were previously precompiled, 7 of them were annotated as known Polycomb Proteins (Cbx5, Eed, Ezh2, Pcgf1, Pcgf2, Rbbp7, Suz12).

The ranking employed in our analysis (Supplementary Data 9) is calculated by considering: (1) *cat*RAPID normalized propensity[41]: *z*-score values between −4 and 4 are mapped to [0, 1] range (*z*-score values under −4 are assigned 0, those above 4 are assigned 1). (2) RBP propensity: a measure of the propensity of the protein to bind RNA. It equals 1 if the protein is in the precompiled RBP library or it is similar to one of such RBPs (otherwise, it is set to *cat*RAPID signature overall score)[21]; (3) known RNA-binding motifs: 0 if no RBP-specific RNA motif is found on the RNA sequence, 0.5 if only one of such motif occurrences is found, 1 if multiple motif occurrences are found[42] After summing these values, the ranking score is scaled to [0, 1] range.

The rG4 sequences on L-1 transcripts were predicted using pqsfinder[43] with a score threshold of 45. The cumulative density plots were obtained by binning the position of the predicted feature using a 50 bp window, and they are shown as a relative position. The statistical significance of the association between predicted rG4 sequences and protein binding sites was assessed using the mergePeaks function within HOMER suite (Hypergeometric Optimization of Motif EnRichment)[44]. The association between *cat*RAPID sites and rG4 on the negative strand was shown as a negative control.

## Bioinformatics analysis

RNA-Seq libraries of in-vitro cortical cells transduced with AAV vectors was made using the TruSeq RNA Library kit v2 (Illumina) and run on a NovaSeq sequencer (DRAGEN Germline Pipeline v3). Paired-end reads (2x50bp) were aligned onto the mm10 mouse build using STAR aligner (version 2.7.3a). Gene expression was quantified with featureCounts in stranded mode (Subread version 2.0.0) and the mouse Gencode GTF annotation (version M25). Filtering of raw counts was done using a cutoff of 2 CPM (counts per million) in at least ¾ of the samples. Distances between samples were calculated based on regularized log transformed counts (rlog function in DESeq2 version 1.30.1). Distances were transformed into a dissimilarity matrix by classical Multidimensional Scaling (MDS, cmdscale in R version 4.0.5) to evaluate the reproducibility of the RNA-Seq experiments. Differential gene expression (DGE) analysis was made with limma (version 3.46.0) using a linear model incorporating the biological groups and the sequencing runs to control for the additional run effect.

RNA-Seq libraries from FAC-sorted GFP+ cells were sequenced in paired-end mode (2x150bp). Reads were counted with featureCounts in unstranded mode and DGE analysis was made using the RankComp V2 algorithm (REOA version 0.1) to account for the pooled replicates of the in-utero experiment and controlling the false discovery rate at 0.05 (Benjamini-Hochberg's method). Heatmaps were rendered using pheatmap (version 1.0.12) and annotated with ComplexHeatmap (version 2.6.2).

Gene ontology (GO) enrichment analyses have been performed using gprofiler2[45,46] querying the GO biological process (GO:BP) database only and limiting the gene set used as background (i.e., universe) to the set of expressed genes. Default hypergeometric test has been used for significance testing. Correction for multiple testing has been performed by using the g:SCS algorithm as suggested in[14]. Significant were considered those GO terms scoring an adjusted *p*-value < 0.1 and associated to more than 2 and 5 significant genes, when the total number of significant genes is in the order of $10^2$ and $10^3$, respectively).

To test whether the shL1s up- and down-regulated genes resulting from our in vivo and in vitro differentially expressed analyses were enriched in given gene sets of interest, custom R script has been used. First, the number of genes in common between our up/downregulated genes and the gene set of interest has been calculated. Next, the same calculation has been computed replacing the up/downregulated genes with an equal number of genes randomly selected from the set of expressed genes, for 1000 times. *Z*-score values have been computed and next converted to *p*-values (*pnorm* R base function). Finally, the so-obtained *p*-values have been corrected by using Benjamini-Hochberg (BH) false discovery rate (FDR). Threshold for significance has been set to 0.05.

To identify potential off-target transcripts recognised by the shL1a and shL1b assays, the nucleotide sequences of the two assays were aligned to the mm10 reference transcriptome (gencode vM25) by using bowtie (v1.2.3) allowing 0 mismatches and selecting only end-to-end matches, as of bowtie default (parameters: -f -S -y -a -v 0). To each off-target transcripts was then associated the expression level of the corresponding genes that were previously classified as "expressed" or "non-expressed" based on the threshold of expression previously used to a priori include/discard genes from the DE analysis[47].

The inference of transcriptional regulators for the in vitro and in utero experiments was made with Lisa (version 2.2.4 under Python 3.8), after selecting genes with an absolute log2 fold change of at least 0.2. The Lisa regulatory scores were based on public Cistrome DB ChIP-Seq experiments[48,49], after subtracting a background score obtained from 3000 background genes.

Additional evidence for Lisa inferred transcriptional regulators was obtained from ChIP-Seq data in the ChIP-Atlas database (https://chip-atlas.org), using the Enrichment Analysis module to query mouse experiments with MACS2 ChIP-Seq peaks with threshold for significance of 100 and located at a maximum distance of 1 kb from annotated TSS.

The quantification of reads mapping to different L1 subfamilies in the shL1-a and shL1-b in-vitro experiment was done using TEspeX (version 1.0.3)[50], a tool that quantifies the expression of TEs avoiding counting sequencing reads deriving from exonized TE fragments embedded in canonical transcripts, in the reverse strand option to account for Illumina TruSeq strand orientation. For the in-vivo dataset, TEspeX was run in the unstranded mode.

Single-cell Drop-seq data from P0 (birth) C57BL/6 J brain cortices were obtained from[51]. The quality control (QC) metrics were done using scater (version 1.14.6) and single cells with a high mitochondrial content, or low library size or low number of expressed genes, were removed from the single-cell analysis. The genes with less than 0.005 counts on average among the 7111 single cells were filtered-out and normalization was done with scran (version 1.14.6). The normalized data was matched to the cell-type annotation originally performed by

the authors and used for the deconvolution of bulk RNA-Seq data. The percentage of cell types in the in-vitro model was estimated by MuSiC (version 0.1.1). The four cell types associated with a change in their estimated cell percentages ($p < 0.05$, ANOVA test), were further analyzed for putative cell markers using Signac (version 0.0.9) and Seurat (version 4.0.2). The TSNE plots were generated from the top 20% highly variable genes (HVGs) (scran), after retention of the first 9 principal components of the principal components analysis (PCA) (scran) and t-SNE on the reduced dimensionality using a perplexity of 30 (scater).

ChIP-Seq libraries of in-vitro cortical cells transduced with AAV expressing shL1-a ware made using the ThruPLEX DNA-Seq kit (Takara) and run on a NovaSeq sequencer.

The H3K27me3 and Ezh2 ChIP-Seq reads corresponding to the shL1 and shCtrl samples were aligned to the mm10 mouse genome using the Bowtie2 aligner. The unmapped reads, not primary aligned reads, and reads aligned with a MAPQ quality score below 30 were filtered out with samtools. The duplicated reads were marked with Picard, then removed and the reads mapped as proper pairs were sorted into a BAM file. The BAM files were transformed into bigWig with deepTools for visualization purposes in the UCSC genome browser and into BEDPE format with bedtools. The metagene analysis was performed using SeqCode, to directly visualize the distribution of the aligned reads present in the BAM files onto gene models. The metagenes were defined by the mm10 RefGene database and the plot was made after counting the number of reads along the region of each gene and averaging this number by the number of genes and the total number of mapped reads. A flanking region of 3 kbs upstream and downstream of the gene body was also considered during the counting. The peak calling was done with epic2, a re-implementation of SICER, on the BEDPE files using matched input samples as control and a FDR threshold of 0.01 when calling the peaks or islands. The differentially bound sites obtained during the epic2 calls were found using DiffBind and a FDR of 0.05. The peaks present in the mm10 ENCODE blacklist regions and the greylists calculated from the samples themselves, were removed and only the replicated peaks common to a specific condition (i.e two out of 2 samples) were considered. The union of peaks from both experimental conditions (i.e shL1 and shCtrl) were used to calculate a binding matrix with scores based on reads counts for each sample. The data was normalized by default, based on sequencing depth and the differential analysis was also performed by default, using DESeq2 and a FDR threshold of 0.05. The genes present in each genomic region found to be differentially bound by Ezh2 or differentially methylated, was obtained from the mm10 UCSC knownGene set and overlapped with genes found to be differentially expressed by RNA-Seq. The intersection between H3K27me3 and Ezh2 ChIP-Seq data was obtained using bedtools and a FDR of 0.05 for both H3K27me3 and Ezh2 differential peaks.

## Statistical analysis

Statistical analyses were performed using GraphPad Prism 7 according to the number of replicates and group design. In the figure legends, statistical tests used and the nature and numbers of samples analyzed (defined as *n*) are reported. Sample sizes were based on published experiments and previous experience in which differences were observed. No power calculations or statistical test to pre-determine the sample size were used. For RNA-Seq experiments, the statistical treatments were done with R version 4.0.3 and Bioconductor release 3.12, as described under those sections. For ChIP-Seq experiments, R version 4.2 and Bioconductor release 3.16 were used.

## Reporting summary

Further information on research design is available in the Nature Portfolio Reporting Summary linked to this article.

## Data availability

All data and materials supporting the findings of this study are available in the main text or the Supplementary Information, and from the corresponding authors upon request. The raw RNA-Seq and ChIP-seq data have been deposited at ENA (European Nucleotide Archive) under the series accession codes PRJEB48280, PRJEB48281 and PRJEB58556. Sequence alignment was done using the mouse reference genome GRCm38 [https://www.ncbi.nlm.nih.gov/assembly/GCF_000001635.20/]. Inference of transcriptional regulators was done using public ChIP-Seq data form Cistrome DB (http://cistrome.org/db/#/) and ChIP-Atlas database (https://chip-atlas.org). Source data are provided with this paper.

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

## Acknowledgements

We are indebted to all the members of the S.G., R.S. and G.G.T. laboratories and of the RNA Initiative@IIT community for thought-provoking discussions. We are grateful to technical and administrative staff of IIT, especially to Eva Ferri and Omar Peruzzo. A.S. is supported by the European Union's Horizon 2020 research and innovation program under the Marie Skłodowska-Curie grant agreement No 754490 within the MINDED project. We gratefully acknowledge Alessandro Parodi, the HPC infrastructure and the support team at Fondazione Istituto Italiano di Tecnologia. The project was funded with intramural IIT (S.G.) funding, intramural SISSA (R.S.) funding and with the Project "National Center for Gene Therapy and Drugbased on RNA Technology" (CN00000041). Financed by NextGenerationEU PNRR MUR – M4C2 – Action 1.4- Call "Potenziamento strutture di ricerca e di campioni nazionali di R&S" (CUP J33C22001130001) (S.G.).

## Author contributions

Conceptualization: D.M., R.S., S.G. Methodology: D.M., A.S., P.L., A.A., F.A., A.C., L.F., V.D.C., D.V. Investigation: D.M., A.S., P.L., R.S., S.G. Visualization: D.M., A.S., P.L., A.A., F.A., L.P. Funding acquisition: R.S., S.G. Project administration: D.M., R.S., S.G. Supervision: L.P., G.G.T., R.S., S.G. Writing – original draft: D.M., R.S., S.G. Writing – review & editing: D.M., D.D., F.P., C.S., L.P., G.G.T., R.S., S.G.

## Competing interests

The authors declare no competing interests.
