## [Peer Review File · Nature Communications]

LINE-1 regulates cortical development by acting as long non-coding RNAsREVIEWER COMMENTS

Reviewer #1 (Remarks to the Author):

Comments

General

This study addresses a very important issue which is the role of L1 expression in the developing nervous system, including at stages when most neurons are at a postmitotic stage. The claim that L1 mRNAs act primarily as non-coding long RNAs is well supported by the experiments, even though it would have been useful to verify the role the encoded reverse transcriptase (RT) and endonuclease proteins. To that end, gain of function experiments with wild type and mutated versions of the two enzymatic activities might have been interesting (also the use of RT inhibitors).

Indeed, although transcriptional repression can be explained by the mechanism proposed by the authors, activation of gene expression is a bit more difficult to integrate in the model. It is not impossible that repression and activation by L1 are regulated by different mechanisms, including the formation of nicks (endonuclease activity) to activate transcription (topoisomerase-like activity).

The authors use in vivo and in vitro models. The comparison between these two models is very challenging as, beyond differences between in vitro and in vivo environments, ages are different and the tools allowing for shRNA expression (transfection in vivo and AAV in vitro) do not allow a sensible comparison of the results. Thus, the only (extremely important) point that remains is the interaction with PRC2 proteins.

Finally, in vitro and in vivo, it would have been useful to identify the cells in which (i) L1 is down-regulated and (ii) show differences in gene expression (ISH) and in the distribution of the epigenetic marks (IHC).

All these comments that will be better illustrated below do not modify the opinion that this is a timely paper which, if the comments are taken into consideration, at an experimental and/or editorial level, deserves publication as many neurobiologists will benefit from this rather original understanding of a role of genetic mobile elements in brain development and physiology. A role non necessitating a classical “jumping activity” which has been reported in dividing progenitors but is very unlikely in postmitotic neurons where L1 transcripts and proteins continue to be expressed.

By the page/figure

Introduction

Even non-full-length transcript can be active.

Activity can include RT and DNA nicks with the endonuclease. In fact, such nicks can enhance transcription through a topoisomerase-like effect on the chromatin structure (especially for long genes).

Among the repressive mechanisms is the expression of MIWIs.

Somatic L1 retro-transposition is important but may not be fully related with the control of cell division or differentiation.

Hypothesis driven study (lines 54-55), which is fine.

L1 silencing in vivo

Electroporation at E12.5, analysis at E13.5, E14.5 and E18.5. Silencing is not complete (around 50%), but is it complete in transfected cells? It might be useful to know the percentage of transfected cells (GFP-expressing). Another important point, not totally clear, is the homogeneity of the transfection throughout the cortex. The calculations of Fig.1d, f and h are convincing but it would be useful to have enlargements showing the double labeling between the different markers and GFP+ cells, used for the calculations. Impaired migration is an important information (Fig1i), but the statistical significance should be better explained. In spite of these remarks, the data support the conclusion that L1 RNAs maintain the correct pool of radial glial cells and it is likely, if the statistical analysis is correct (this needs an answer from the authors), that they “play a role in the ability of newborn neurons to correctly migrate along the cortical plate”.

The RNA-seq analysis that follows is important with genes upregulated and others downregulated, even though it is impossible to say whether this corresponds to a direct regulation of transcription. In case of gene repression, experiments described later in the manuscript suggest that repressive histones are recruited, but nothing is said concerning CpG methylations (only in one very specific case) or hybridization with L1 truncated or full-length sequences often present in non-coding regions, in particular within introns belonging to regulated genes. Activation could result from the inhibition of inhibitory pathways, but could also be a consequence of the expression of ORF2p and of its endonuclease activity, in particular at the level of long-genes which are heavily expressed in the nervous system. The

reviewer understands that this type of investigation is difficult to do in vivo, but it might be useful to discuss these possibilities at the end of the manuscript.

The comparison with mESCs provided by the authors is a bit difficult to understand, if not to propose that L1 RNA silencing down-regulates, both in the brain and in the stem cells, genes associated with high metabolism, thus rapid growth. However, high metabolism is also a characteristic of differentiated neurons and one wonders whether the comparison is so useful. This does not invalidate the suggestion that L1 RNAs (or their never mentioned protein products) regulate the balance between proliferation and differentiation in neural progenitors, but the statement is so general that it would be quite hard to refute.

Finally, for this chapter but also for the experiments described in the rest of the study, gain of function experiments are not provided. Indeed, loss of function experiments are more telling, but this choice should be justified.

L1 silencing in vitro

This chapter is of interest but it is clear that, in addition with differences inherent to in vitro conditions, the data obtained cannot be compared with the in vivo ones. Indeed, the cells are dissociated from E17.5 embryos and L1RNA down-regulation is obtained with an AAV, a technology that requires a few days before shRNA expression (would be nice to follow the expression). This does not mean that the approach is not interesting but may cast doubt on the coherence of the manuscript, beyond the idea, supported by the data, that L1 RNAs act as regulatory non-coding RNA, not precluding also an effect of the translation products as mentioned above.

Extended data Figure 2 clearly demonstrates that L1 is expressed during differentiation (possibly not only at the RNA level) and established a correlation with the deposition of H3K9me3 (ChIP experiment). Change in the methylation of CpG site 9 of L1MdTf promoter is nicely demonstrated, but one wonders about the other L1 families and the comment (line 126) on YY1 binding seems rather circumstantial.

Yet, the claims on gene regulation (up and down) are supported by the data, as well as the effect of the inhibition of L1 expression on the balance between cell types. The increase in upper layer neurons is at odds with the in vivo experiment, but this is not really surprising given the radical differences between the two models. This makes the invitro/in vivo comparison (lines 189-199) possibly unnecessary. Unless making is more complete by TAD

identification (ATAC seq).

It remains that the major conclusion of a possible effect of L1 RNA down-regulation (shRNA strategy) on chromatin remodeling in vitro and in vivo is valid. The verification of this hypothesis developed in the rest of the manuscript may have benefited from some IHC experiments showing changes in the patterns (in vitro and in vivo) of several epigenetic marks, including repressive histones and MeCP2 staining.

PRC2 proteins

The rest of the manuscript on the physical and functional interactions between PRC2 and L1 RNAs is convincing. The modeling analysis (catRAPID, G4 structure) and the fact that a large majority of the L1 transcripts are associated with the chromatin and specifically pulled-down by Suz12 and Ezh2 antibodies is important and supports the non-coding RNA hypothesis. The correlation between upregulation and lesser Ezh2 and H3K27me3 binding (the opposite for down-regulation) is also in support of the model, even though it could be seen, once more, as circumstantial evidence. Here again, the model would be comforted by L1 gain of function experiments.

Reviewer #2 (Remarks to the Author):

Mangoni et al. developed a system for the knockdown of transcripts of L1 elements that, unsurprisingly, lead to impaired brain development and to global changes in transcription programs (Fig 1 and 2). Then, the authors try to link these global changes to the hypothesis that L1 transcripts regulate PRC2 (Fig 3 and 4). The first part of the work (Fig 1 and 2) does not seem to be sufficiently advancing the field to justify a publication in Nature Comms on its own. The second part of the work (Fig 3 and 4) cannot directly support a model for the regulation of PRC2 by L1, with the basic perturbation experiments cannot dissect direct from indirect effects on PRC2 and transcriptional regulation in general. As a whole, the work does not provide a mechanism and even the general hypothesis regarding the regulation of PRC2 by L1 transcripts is not directly supported.

Specificity, the authors demonstrated that L1 is important for the embryonic development

of mouse neurons in vivo (Fig 1) and in vitro (Fig 2) using siRNA knockdown. This part is reasonably well supported by imaging and genomic analysis (mainly RNA-seq), even though the bioinformatics analysis and its description across the text are not pointed and confusing at times. Yet, these findings seem rather incremental, with a role for L1 in neural development has been known for over 20 years (e.g. see review by Kamiguchi et al 1998 [PMID: 9770339] for an early discussion into this topic). Then, the authors attempt to link changes in transcription to the histone methyltransferase polycomb repressive complex 2 (PRC2). They bring PRC2 into the picture in two ways: (i) first, by analysing publicly available data and using it to identify “transcription factor binding sites” for PRC2 subunits (Fig 3b). Next, (ii) they show that PRC2 subunits bind to L1 RNA using a basic RNA-IP experiment. Yet, both these observations are expected: (1) PRC2 is a master regulator of transcription factors in mouse development [PMID: 31123062], including neural development [PMID: 25367430], and PRC2 binds promiscuously to thousands of transcripts in cells [PMID: 24077223]. There are other observations in Fig 3, but none of them can DIRECTLY link the hypothesised L1→PRC2 pathway to observations in Fig 1 and 2. In fact, some of the observations in Fig 3 are simply predictions, not experimental results. Figure 4 simply demonstrates that the recruitment of PRC2 to chromatin and the deposition of the repressive H3K27me3 mark altered upon L1 knockdown, but this could be either the result of a direct effect or (more likely) indirect global effects related to impaired development and/or nuclear structure.

I included below a few comments for the authors, but I don't see how they could address them in a reasonable time given the preliminary shape of this manuscript. In its current shape, this work did not establish a direct functional link between L1 to PRC2, beyond the mere physical interactions, which are not surprising. Unfortunately, I have to recommend for a rejection.

Major points

1. Fig 1 and 2: Fig 1 and 2 represent mainly RNA expression data and some basic CHIP-qPCR experiments that demonstrate the involvement of L1 elements in neurodevelopment. The involvement of L1 in neurodevelopment is not new (e.g. see Kamiguchi et al 1998 [DOI:

10.1006/mcne.1998.0702]), and the data in Fig 1, 2 does not provide any evidence that could dissect global effects. Accordingly, L1 knockdown could have affected neurodevelopment indirectly (e.g. through a nuclear organisation or metabolism), rather than through the direct involvement of the regulation of histones modifications and transcriptional regulation. Therefore, it would be appropriate to moderate the bold statement from L200 (“In summary, L1 RNAs regulate chromatin remodelling and transcriptional control”). This statement is not directly supported by the data in this manuscript. More importantly, it is important to discuss in what way observations in Fig 1 add on top of previous works.

2. Fig 1b and 2c: The data in Figures 1b and 2c demonstrate a 2-fold reduction in the L1 expression level, at best. This is a key limitation of the experimental system. If knockdown efficiency cannot be improved then the authors should at least discuss this limitation textually.

3. Fig 3f end ED 5e-f: The catRAPID analysis is merely predictive and misleading without experimental validation. The authors should either validate these predictions experimentally or remove them from the manuscript.

4. L332: “These data suggest that L1 RNAs may exert ‘sponge’ or ‘guide’-like mechanisms in regulating PRC2 recruitment to target sites and deposition of repressive epigenetic marks.” This statement is the closest thing for a model that was put forward in this manuscript, but it is a rather vague and yet a long stretch away from the data. The authors should set a hypothesis (or hypotheses) regarding a working model to explain how L1 regulates PRC2 and then set up to test it experimentally. To support their hypothesis, the authors would have to develop a significant line of research that would extend far beyond the preliminary data in Fig 3-4. First, they will have to experimentally identify how PRC2 and L1 RNA interact. This would require answering the question: what bases in L1 RNA are sufficient and required for binding to PRC2? Based on this investigation, the authors should be able to introduce point mutations to L1, that would disrupt its interactions with PRC2. If it is practical to introduce these mutations to selected L1 elements in cells then it could be used in conjugation with ChIP-seq and RNA-seq to show that the recruitment and activity of PRC2 at the mutated

locus has changed according to the hypothesis. Do L1 RNA regulate the activity of PRC2 at their site of transcription (i.e. in cis) or other loci (i.e. in trans)? This is a question that should be considered while setting up a model and designing experiments to test it. If the authors believe that the effect is rather global (e.g. global 'sponge') then they should identify the L1 region that is sufficient for binding to PRC2 and they should overexpress it with the expectation that it could rescue the PRC2-linked global effects seen after by L1 knockdown. The same experiments can then be carried out in the presence or absence of PRC2 inhibitor or knockout, with the expected result of epistasis in case the transcriptional changes are attributed to the hypothesized L1—>PRC2 pathway.

5. Where in the nucleus the siRNA knockdown of L1 takes place? Is this type of perturbation affect the abundance of L1 RNA that is associated with facultative heterochromatin or is it take place in the nucleoplasm? Or co-transcriptionally? This question seems relevant and needs to be addressed experimentally as if the knockdown does not take place at the site of transcription it would exclude regulation in cis.

Minor points

6. L49: The text says that "Ezh2 is the catalytic enzyme", but PRC2 is the enzyme while EZH2 is the catalytic subunit.

7. Extended Data Fig.3: it is impossible to read the text within ED Fig 3b, which makes all the TSNE representations across ED Fig 3e-h obsolete.

Reviewer #4 (Remarks to the Author):

Summary of the article

Previously Percharde et al. has shown that the transposon LINE1 (L1) expression is essential for embryonic stem cell self renewal in mouse, by acting as a nuclear scaffold RNA that activates ribosomal RNA (rRNA) synthesis and inhibits Dux, the master activator of a

transcriptional program specific to the 2-cell embryo (PMID: 29937225). In addition to its expression in the pre-implantation embryos, L1 is also expressed in neuro-progenitor cells (NPCs) and the active copies can lead to somatic retrotranspositions and generate novel mobile element insertion (MEI) mutations. In this manuscript, Mangoni et al. explored the regulatory roles of L1 RNA in the neuro-progenitor cells, and discovered that:

- 1) L1 RNA can serve as regulatory non-coding RNA that contribute to the (A) differentiation of NPCs and (B) migration of the post-mitotic neurons.
- 2) The regulatory role of L1 RNA is through interacting with the subunits of the protein complex PRC2, and regulating its targets for histone methylation (H3K27me3), a marker for silenced chromatin.

The insight of L1 RNA serving as regulatory components, affecting the chromatin structure and contributing to the NPC differentiation is very exciting, however, a few key aspects of the supporting evidence need to be further clarified to strengthen the biological findings.

Major critiques

- 1) Systematic bias from the RNA interfering probes -- shL1-a and shL1-b

While the efficacy of shL1-a and shL1-b to knock down L1 expression has been shown in Fig. 1 and Fig. 2, their possible off-target effect needs to be evaluated with more details. L1 sequences are abundant in the mouse genome and many of them have been exonized into coding sequences, for instance, it was estimated that 100 – 300 L1 elements overlap with genes in the mouse genome (PMID: 17594509). Thus, it is very possible that the interfering RNA probes could knock down mRNAs of certain genes. This effect can be easily evaluated by looking for L1 fragments, particularly the shL1-a or shL1-b sequences, among the coding sequences of the top down-regulated genes.

In addition, the authors used two small RNAs, shL1-a and shL1-b, that target the similar regions in L1 as probes for RNA interference the in vitro experiment. While both shL1-a and shL1-b could downregulate L1 RNAs across different subfamilies and led to a large overlap of DEGs, it is still surprising to observe that across the whole transcriptome the shL1-b experiments still clustered much closer to the controls rather than the shL1-a experiments (Extended figure 3c). While this difference could be explained by truncated L1 RNAs that could only be interfered with either shL1-a or shL1-b, it remains possible the bias could be contributed by the difference in the off-targets.

Finally, despite the difference between shL1-a and shL1-b in the in vitro experiment, the authors only applied shL1-a in the in vivo experiment, and observed a significant number of genes that were regulated in opposite patterns from the in vitro experiment (line 191). To rule out the systematic bias introduced by the selection of the interfering probe, it is therefore essential to replicate the in vivo experiment with the shL1-b probe and reevaluate the comparison with the in vitro experiments.

2) The function of L1 RNA through interacting with PRC2 and modifying histone methylation
In the second half of the manuscript, the authors provided supporting evidence for a model where L1 RNAs interact with protein Suz12 and Ezh2 and regulate the histone methylation activities (H3K27me3) of the PRC2 complex. They showed that the localization of L1 RNAs are significantly associated with chromatin, and a significant amount of L1 RNAs can be co-IPed with both Suz12 and Ezh2, when using a house keeping gene (GAPDH) and 45s rRNA as controls. Since Suz12 and Ezh2 are primarily localized in the chromatin, this co-IP enrichment of L1 RNAs could be due to the localization of the RNAs, instead of direct interactions with the suggested proteins. It is therefore essential to replicate the co-IP results using the chromatin fraction of RNAs (instead of the whole cell lysate), or using other control RNAs with similar localization preference but has no interactions with the PRC2 complex.

The authors then investigated the H3K27me3 statuses of the DEGs in the L1-knock down experiments, and showed that in a panel of 16 (8 upregulated and 8 downregulated) genes, the changes of H3K27me3 and the Ezh2 binding were consistent with the changes of gene expression. While this is a very strong evidence supporting the proposed mechanism of the L1 RNA, 16 genes remain a small subset of all DEGs. It will be important to demonstrate how representative these 16 genes are and if the pattern can be extrapolated towards other genes. Specifically, how did these genes were selected in the first place, and what about the changes in other DEGs? More robust evidence can be provided with ChIP-seq analyses to pull down Ezh2 and H3K27me3 marks in the same L1-knock down and control cell cultures, just like their ChIP-qPCR experiment, but to investigate all genes instead of a limited gene panel.

Minor critiques

1) It is essential to show the exact p values (e.g., $p=1.1e-07$, line 170), instead of the range of the p values (e.g., $p < 0.01$, line 69). In the in vivo experiments (line 68-81), it is also unclear how many different comparison tests were performed, and whether the p values were corrected for the number of comparisons (e.g., multiple testing correction).

2) The ChIP-qPCR experiment (Figure 4) lacks proper negative controls such as genes with no expression changes in the L1-knock down culture.

Reviewer #1:

Comments

General

This study addresses a very important issue which is the role of L1 expression in the developing nervous system, including at stages when most neurons are at a postmitotic stage. The claim that L1 mRNAs act primarily as non-coding long RNAs is well supported by the experiments, even though it would have been useful to verify the role the encoded reverse transcriptase (RT) and endonuclease proteins. To that end, gain of function experiments with wild type and mutated versions of the two enzymatic activities might have been interesting (also the use of RT inhibitors).

Indeed, although transcriptional repression can be explained by the mechanism proposed by the authors, activation of gene expression is a bit more difficult to integrate in the model. It is not impossible that repression and activation by L1 are regulated by different mechanisms, including the formation of nicks (endonuclease activity) to activate transcription (topoisomerase-like activity).

The authors use in vivo and in vitro models. The comparison between these two models is very challenging as, beyond differences between in vitro and in vivo environments, ages are different and the tools allowing for shRNA expression (transfection in vivo and AAV in vitro) do not allow a sensible comparison of the results. Thus, the only (extremely important) point that remains is the interaction with PRC2 proteins.

Finally, in vitro and in vivo, it would have been useful to identify the cells in which (i) L1 is down-regulated and (ii) show differences in gene expression (ISH) and in the distribution of the epigenetic marks (IHC).

All these comments that will be better illustrated below do not modify the opinion that this is a timely paper which, if the comments are taken into consideration, at an experimental and/or editorial level, deserves publication as many neurobiologists will benefit from this rather original understanding of a role of genetic mobile elements in brain development and physiology. A role non necessitating a classical “jumping activity” which has been reported in dividing progenitors but is very unlikely in postmitotic neurons where L1 transcripts and proteins continue to be expressed.

We thank very much the Reviewer for his/her positive comments on the timing and value of the paper. We are also grateful for his/her suggestions to improve the manuscript.

By the page/figure

Introduction

Even non-full-length transcript can be active.

We thank the Reviewer for spotting out this oversight. We added to the text the following sentence:

“and to foster mobilization of truncated L1s”

Activity can include RT and DNA nicks with the endonuclease. In fact, such nicks can enhance transcription through a topoisomerase-like effect on the chromatin structure (especially for long genes).

We thank the Reviewer for this important observation. We added to the text the following sentence: “Uncontrolled expression of LINE-1 triggers oxidative stress-induced DNA strand breaks leading to cell death and neurodegeneration” followed by the appropriate reference.

Among the repressive mechanisms is the expression of MIWIs.

We thank the Reviewer for spotting out this oversight. We added to the text the following sentence: “affecting stability of L1 transcripts through the activity of MIWI/piRNA pathway and RNA helicases”

Somatic L1 retro-transposition is important but may not be fully related with the control of cell division or differentiation.

We thank the Referee for this observation.

Hypothesis driven study (lines 54-55), which is fine.

L1 silencing in vivo

Electroporation at E12.5, analysis at E13.5, E14.5 and E18.5. Silencing is not complete (around 50%), but is it complete in transfected cells? It might be useful to know the percentage of transfected cells (GFP expressing).

We thank the Reviewer for the observation and the possibility it gave us to better clarify this point. At E14.5 GFP⁺ cells are $18,7 \pm 5,1$ % and $16,4 \pm 4,5$ % of total cells in the neocortex (DAPI⁺), for shCtrl and shL1-a, respectively. In Fig. 1b we show that the efficiency of L1 knockdown (for L1MdA, L1MdGf and L1MdTf) in GFP⁺ cells FACS-sorted from E14.5 neocortex of mice i.u.e. with shL1-a is about 50-60% compared to GFP⁺ cells from control mice (i.u.e. with shCtrl). So, even in transfected cells the efficiency of the silencing is not complete. Unfortunately, we were never able to achieve better results with shRNAs at our disposal. In this regard, we suspect that L1 RNAs are difficult targets to silence by RNA interference approaches because of their heterogeneity and localization (mostly nuclear, and chromatin-associated). Moreover, interactions with chromatin and proteins may entail the formation of secondary structures and mask sequences targeted by shRNAs.

Another important point, not totally clear, is the homogeneity of the transfection throughout the cortex. The calculations of Fig.1d, f and h are convincing but it would be useful to have enlargements showing the double labeling between the different markers and GFP⁺ cells, used for the calculations.

We thank the Reviewer for the observation and the possibility it gave us to provide additional information.

Enlargements have been added for each image in the figures showing the double labeling between the specific marker and GFP. For cell counts, double positive cells (GFP⁺/marker⁺) were counted and normalized on the total number of transfected (GFP⁺) cells.

Impaired migration is an important information (Fig1i), but the statistical significance should be better explained. In spite of these remarks, the data support the conclusion that L1 RNAs maintain the correct pool of radial glial cells and it is likely, if the statistical analysis is correct (this needs an answer from the authors), that they “play a role in the ability of newborn neurons to correctly migrate along the cortical plate”.

We thank very much the Reviewer for suggesting these important improvements.

To strengthen our conclusions about the effect of L1 silencing upon proliferation and commitment of neuronal progenitors, we repeated i.u.e. experiments with the second shRNA against L1s (shL1-b) at E14.5 and we found comparable results for Pax6 and NeuroD1 with the first shRNA used (shL1-a). In fact, using both the shL1s the number of Pax6⁺ cells is decreased and NeuroD1⁺ cells increased at E14.5 (Fig. 1c-d and Extended Data Fig. 1b-c). Therefore, we are confident with our conclusion that L1 RNAs maintain the correct pool of radial glial cells, while L1 RNAs knockdown impairs proliferation of progenitor cells by promoting neuronal commitment.

To strengthen our conclusions about the effect of L1 silencing upon migration of neuronal cells at E18.5, we increased the number of experiments. In this revised version of the manuscript, with n=4 for shCtrl and n=3 for shL1-a i.u.e. experiments, we achieved a significant increase of GFP⁺ cells in bins 6 (P<0.01) and 7 (P<0.05) when L1 RNAs are silenced compared to control condition. Thus, we confirmed that L1 RNAs have a role in regulating the correct migration of newborn neurons along the cortical plate, while the down-regulation of L1 RNAs promotes the accumulation of cells in the deep layers of the cortex, particularly through layers 6 and 5 (Fig1 g,h).

The RNA-seq analysis that follows is important with genes upregulated and others downregulated, even though it is impossible to say whether this corresponds to a direct regulation of transcription. In case of gene repression, experiments described later in the manuscript suggest that repressive histones are recruited, but nothing is said concerning CpG methylations (only in one very specific case) or hybridization with L1 truncated or full-length sequences often present in non-coding regions, in particular within introns belonging to regulated genes. Activation could result from the inhibition of inhibitory pathways, but could also be a consequence of the expression of ORF2p and of its endonuclease activity, in particular at the level of long-genes which are heavily expressed in the nervous system. The Reviewer understands that this type of investigation is difficult to do in vivo, but it might be useful to discuss these possibilities at the end of the manuscript.

We thank the Reviewer for his/her observation. We added to the Discussion section the following sentence:

L1s may exert these regulatory functions through several mechanisms. While the role of L1s mobilization is unlikely given the lack of substantial consequences of RT inhibition, the synthesis of ORF2p could enhance transcription of pro-neural genes through a topoisomerase-like effect on the chromatin structure by its endonuclease activity (12).

The comparison with mESCs provided by the authors is a bit difficult to understand, if not to propose that L1 RNA silencing down-regulates, both in the brain and in the stem cells, genes associated with high metabolism, thus rapid growth. However, high metabolism is also a characteristic of differentiated neurons and one wonders whether the comparison is so useful. This does not invalidate the suggestion that L1 RNAs (or their never mentioned protein products) regulate the balance between proliferation and differentiation in neural progenitors, but the statement is so general that it would be quite hard to refute.

We thank the Reviewer for this comment. The comparison between L1 RNA knockdown in embryonic stem cells and developing brain cortex showed that many genes involved in ribosome biogenesis and translation are down-regulated as a consequence of L1 silencing. This suggests that the impact upon cellular metabolism through the impairment of ribosomal activity may be a “core effect” of the decreased levels of L1 transcripts even in different biological systems (ESCs and NPCs), although both characterized by mitotic activity. Percharde et al. (PMID: 29937225) demonstrated that interaction of L1 transcripts with Nucleolin/Kap1 complex inhibits Dux-mediated 2-cell embryo transcriptional program and promotes ribosomal RNA transcription, which contributes to maintaining embryonic cell’s proliferation and self-renewal. In our work, we found that ribosome biogenesis and proliferation of neural progenitors are both affected by L1 silencing. We did not investigate in depth the possible link between L1 RNAs and ribosomal disfunctions in relation to the effect upon cell division and neuronal commitment since it was not the major purpose of this work. However, we believe that this evidence could be of interest as a cue for future works aimed at studying the role of L1 expression in regulating cell metabolism in the developing brain and in neurodegenerative diseases characterized by metabolic disfunctions.

Finally, for this chapter but also for the experiments described in the rest of the study, gain of function experiments are not provided. Indeed, loss of function experiments are more telling, but this choice should be justified.

We thank the Reviewer for giving us the chance to better clarify this important point. We agree that gain-of-function experiments could be useful in studying the effects of L1 RNAs in the experimental setting of choice. However, we believe that in this experimental system, overexpressing full length L1 RNAs could lead to at least two potential confounding effects: i. some L1 RNAs functions could be due to their activity in the site of transcription that could not be mimicked by ectopic expression; ii. binding to PRC2 depends on the anatomy of the single L1 RNA, its repertory of binding sites and its location in the genome. Preliminary experiments carried out with Nanopore direct RNA sequencing show that no full length L1s are apparently expressed at 21 div. The complete list of L1 RNAs, their genome mapping and their interplay with RBPs are the topic of the next chapter of this exciting story.

L1 silencing in vitro.

This chapter is of interest but it is clear that, in addition with differences inherent to in vitro conditions, the data obtained cannot be compared with the in vivo ones. Indeed, the cells are dissociated from E17.5 embryos and L1RNA down-regulation is obtained with an AAV, a technology that requires a few days before shRNA expression (would be nice to follow the expression). This does not mean that the approach is not interesting but may cast doubt on the coherence of the manuscript, beyond the idea, supported by the data, that L1 RNAs act as regulatory non-coding RNA, not precluding also an effect of the translation products as mentioned above.

As pointed out by the Reviewer, *in vivo* and *in vitro* effects of L1 knockdown are affected by significant biological differences characterizing the two steps of neuronal development. The comparison is aimed to pinpoint the possibility that L1 transcription may regulate gene expression in spatial- and temporal-specific manner with distinct effects in different stages of the development of the brain, more than necessarily looking for commonalities between the two biological systems.

Extended data Figure 2 clearly demonstrates that L1 is expressed during differentiation (possibly not only at the RNA level) and established a correlation with the deposition of H3K9me3 (ChIP experiment). Change in the methylation of CpG site 9 of L1MdTf promoter is nicely demonstrated, but one wonders about the other L1 families and the comment (line 126) on YY1 binding seems rather circumstantial.

We thank the Reviewer for the observation and the possibility it gave us to better clarify this point. As suggested by the Referee, it would be interesting to look at differences in promoter methylation also for other L1 families in addition to L1MdTf. This information would be important to see if evolutionary different L1 families are characterized by different ability to respond to regulatory pathways and mechanisms which may exert control over L1 expression, particularly in relation to the activity of specific transcription factors, the activity of which is cell-type- and/or time-specific in the development of the brain. However, the aim of the work is not focused on the study of factors which may act by controlling L1 expression, so we did not fully investigate the mechanistic aspects underlying the regulation of expression for all L1 families. More specifically, concerning promoter methylation, we used L1MdTf family as a model to assess the methylation state of the promoters of evolutionary younger L1 elements, as in (PMID: 29567711). That's because, among L1 families, L1MdTf is the most studied and better characterized especially for the presence of regulatory sequences in the promoter region, besides being among the top transcribed L1 elements in the biological systems studied in this work.

We agree that the comment (line 126) on YY1 binding is rather circumstantial and therefore we eliminated it from this new version of the manuscript.

Yet, the claims on gene regulation (up and down) are supported by the data, as well as the effect of the inhibition of L1 expression on the balance between cell types. The increase in upper layer neurons is at odds with the in vivo experiment, but this is not really surprising given the radical differences between the two models. This makes the invitro/in vivo comparison (lines 189-199) possibly unnecessary. Unless making is more complete by TAD identification (ATAC seq).

We thank the Reviewer for his/her comment. In addition to our response to a previous observation about the difficulties in comparing the two model systems, here we note that the increase of upper layer neurons is one of the common features between *in vivo* and *in vitro* experiments. Immunostainings of brain cortices electroporated with shL1-a at E12.5 and collected at E18.5 showed that Cux1-positive neurons were doubled compared to control mice, although more retained in the lower layers of the cortical plate as a consequence of the impairment of the migration ability. On the other hand, L1 silencing in postmitotic cells isolated from E17.5 cortices resulted in a higher number of NeuroD2-positive cells, along with a higher expression of transcription factors characterizing upper layers neurons (NeuroD2, Pou3f1 and Satb2), although these cells were more immature than controls.

It remains that the major conclusion of a possible effect of L1 RNA down-regulation (shRNA strategy) on chromatin remodeling in vitro and in vivo is valid. The verification of this hypothesis developed in the rest of the manuscript may have benefited from some IHC experiments showing changes in the patterns (in vitro and in vivo) of several epigenetic marks, including repressive histones and MeCP2 staining.

We thank the Reviewer for his/her suggestion. IHC experiments will be carefully carried out in a follow-up paper.

PRC2 proteins

The rest of the manuscript on the physical and functional interactions between PRC2 and L1 RNAs is convincing. The modeling analysis (catRAPID, G4 structure) and the fact that a large majority of the L1 transcripts are associated with the chromatin and specifically pulled-down by Suz12 and Ezh2 antibodies is important and supports the non-coding RNA hypothesis. The correlation between upregulation and lesser Ezh2 and H3K27me3 binding (the opposite for down-regulation) is also in support of the model, even though it could be seen, once more, as circumstantial evidence. Here again, the model would be comforted by L1 gain of function experiments.

We thank the Reviewer for finding our results convincing. In the revised version of the paper we have also performed ChIP-seq experiments to evaluate both deposition of H3K27me3 and binding of Ezh2, in a genome-wide scale. First, we observed a significantly higher H3K27 three-methylation upon regulatory regions of down-regulated genes, accordingly with RNA-seq experiments, suggesting that down-regulation of a significant portion of genes after L1 RNAs silencing is due to hypermethylation of their promoter region. No significant changes were observed for genes up-regulated or whose expression didn't change according to RNA-seq. Ezh2 binding was altered by L1 RNAs silencing for 418 genes. The majority of them, 261 (62%; $P = 7.6E-11$), overlapped with H3K27me3 hypermethylated regions (Fig. 4c), suggesting that L1 RNAs knockdown was able to alter, at least for these regions, both Ezh2 occupancy and H3K27me3 deposition. This group of genes are enriched for transcription factors involved in neuronal differentiation. Future work will address the details and functional outputs of PRC2-L1 RNAs interactions at single gene loci and the significance of the difference in the magnitude of variations between H3K27me3 deposition and Ezh2 binding upon L1 RNAs silencing.

Of note, the previous Ab used for ChIP of Ezh2 did not pass the standard of quality controls when used in ChIP-seq experiments. Therefore, in this revised version we decided to eliminate previous data on ChIP experiments on single gene target.

Reviewer #2:

Mangoni et al. developed a system for the knockdown of transcripts of L1 elements that, unsurprisingly, lead to impaired brain development and to global changes in transcription programs (Fig 1 and 2). Then, the authors try to link these global changes to the hypothesis that L1 transcripts regulate PRC2 (Fig 3 and 4). The first part of the work (Fig 1 and 2) does not seem to be sufficiently advancing the field to justify a publication in Nature Comms on its own. The second part of the work (Fig 3 and 4) cannot directly support a model for the regulation of PRC2 by L1, with the basic perturbation experiments cannot dissect direct from indirect effects on PRC2 and transcriptional regulation in general. As a whole, the work does not provide a mechanism and even the general hypothesis regarding the regulation of PRC2 by L1 transcripts is not directly supported.

We thank the Reviewer for her/his comments on our manuscript. However, we respectfully disagree on many of his/her observations.

On:

The first part of the work (Fig 1 and 2) does not seem to be sufficiently advancing the field to justify a publication in Nature Comms on its own.

While the decision to be published in this highly prestigious journal is clearly out of our reach, we respectfully disagree with the statement saying “our study does not seem to be sufficiently advancing the field”. At present, this is the first study that shows the requirement of L1 RNAs for corticogenesis *in vivo*. We consider this a major achievement in the field. We kindly ask the Reviewer to provide us a list of papers he/she considers of reference.

On:

The second part of the work (Fig 3 and 4) cannot directly support a model for the regulation of PRC2 by L1, with the basic perturbation experiments cannot dissect direct from indirect effects on PRC2 and transcriptional regulation in general. As a whole, the work does not provide a mechanism and even the general hypothesis regarding the regulation of PRC2 by L1 transcripts is not directly supported.

We again respectfully but firmly disagree with the Reviewer. In summary we prove that: i. PRC2 binding sites are enriched in genes differentially expressed upon L1 RNA silencing *in vivo* and *in vitro*; ii. PRC2 subunits bind L1 RNAs; iii. silencing L1 RNAs lead to a change in H3K27me3 deposition in differentially regulated genes; iv. silencing L1 RNAs lead to a change in PRC2 binding in a selected group of genes involved in neuronal differentiation and maintenance.

Specificity, the authors demonstrated that L1 is important for the embryonic development of mouse neurons in vivo (Fig 1) and in vitro (Fig 2) using siRNA knockdown. This part is reasonably well supported by imaging and genomic analysis (mainly RNA-seq), even though the bioinformatics analysis and its description across the text are not pointed and confusing at times.

We kindly ask the Reviewer to help us increasing the quality of the manuscript and indicate which part is confusing so that we can improve it.

Yet, these findings seem rather incremental, with a role for L1 in neural development has been known for over 20 years (e.g. see review by Kamiguchi et al 1998 [PMID: 9770339] for an early discussion into this topic).

This is the most confusing part of Reviewer comments. The suggested paper is the following:

Kamiguchi H, Hlavin ML, Lemmon V. Role of L1 in neural development: what the knockouts tell us. *Mol Cell Neurosci.* 1998 Sep;12(1-2):48-55. doi: 10.1006/mcne.1998.0702. PMID: 9770339.

We agree with the Reviewer that the role of L1 in neural development is well known for 20 years. Unfortunately, this paper (and probably the majority of Reviewer comments) refers to L1 as the cell adhesion molecule (L1CAM), and not L1 as the Long Interspersed Nuclear Elements (LINE-1).

Since this paper is quoted twice, we kindly ask the Reviewer to clarify this point, as he/she considers it important.

Then, the authors attempt to link changes in transcription to the histone methyltransferase polycomb repressive complex 2 (PRC2). They bring PRC2 into the picture in two ways: (i) first, by analysing publicly available data and using it to identify “transcription factor binding sites” for PRC2 subunits (Fig 3b). Next, (ii) they show that PRC2 subunits bind to L1 RNA using a basic RNA-IP experiment.

Yet, both these observations are expected: (1) PRC2 is a master regulator of transcription factors in mouse development [PMID: 31123062], including neural development [PMID: 25367430], and PRC2 binds promiscuously to thousands of transcripts in cells [PMID: 24077223]. There are other observations in Fig 3, but none of them can DIRECTLY link the hypothesised L1—>PRC2 pathway to observations in Fig 1 and 2. In fact, some of the observations in Fig 3 are simply predictions, not experimental results. Figure 4 simply demonstrates that the recruitment of PRC2 to chromatin and the deposition of the repressive H3K27me3 mark altered upon L1 knockdown, but this could be either the result of a direct effect or (more likely) indirect global effects related to impaired development and/or nuclear structure.

We thank the Referee for his/her comment.

We cannot discuss what is expected or not in an experiment, it is matter of personal taste.

We agree that some of the observations in Fig. 3 are predictions. We disagree on the “simply” since this is the first time that L1 RNAs are predicted to bind Suz12 and Ezh2. Importantly, predicted bindings are then experimentally proved with the appropriate techniques.

I included below a few comments for the authors, but I don't see how they could address them in a reasonable time given the preliminary shape of this manuscript. In its current shape, this work did not establish a direct functional link between L1 to PRC2, beyond the mere physical interactions, which are not surprising. Unfortunately, I have to recommend for a rejection.

Major points:

1. Fig 1 and 2: Fig 1 and 2 represent mainly RNA expression data and some basic ChIP-qPCR experiments that demonstrate the involvement of L1 elements in neurodevelopment. The involvement of L1 in neurodevelopment is not new (e.g. see Kamiguchi et al 1998 [DOI: 10.1006/mcne.1998.0702]), and the data in Fig 1, 2 does not provide any evidence that could dissect global effects. Accordingly, L1 knockdown could have affected neurodevelopment indirectly (e.g. through a nuclear organisation or metabolism), rather than through the direct involvement of the regulation of histones modifications and transcriptional regulation. Therefore, it would be appropriate to moderate the bold statement from L200 ("In summary, L1 RNAs regulate chromatin remodelling and transcriptional control"). This statement is not directly supported by the data in this manuscript. More importantly, it is important to discuss in what way observations in Fig 1 add on top of previous works.

Again, while we respect Reviewer comments, we are confused about the reference to a paper on the cell adhesion molecule L1CAM.

2. Fig 1b and 2c: The data in Figures 1b and 2c demonstrate a 2-fold reduction in the L1 expression level, at best. This is a key limitation of the experimental system. If knockdown efficiency cannot be improved then the authors should at least discuss this limitation textually.

We thank the Reviewer for his/her comment that helps us to clarify this important point. In Fig. 1b we show that the efficiency of L1 knockdown (for L1MdA, L1MdGf and L1MdTf) in GFP⁺ cells FACS-sorted from E14.5 neocortex of mice i.u.e. with shL1-a is about 50-60% compared to GFP⁺ cells from control mice (i.u.e. with shCtrl). So, even in transfected cells the efficiency of the silencing is not complete. Unfortunately, we were never able to achieve better results with shRNAs at our disposal. In this regard, we suspect that L1 RNAs are difficult targets to silence by RNA interference approaches because of their heterogeneity and localization (mostly nuclear, and chromatin-associated). Moreover, interactions with chromatin and proteins may entail the formation of secondary structures and mask sequences targeted by shRNAs.

We agree with the Reviewer that in the case of silencing the cell adhesion molecule L1, the knockdown efficiency would not be enough.

3. Fig 3f end ED 5e-f: The catRAPID analysis is merely predictive and misleading without experimental validation. The authors should either validate these predictions experimentally or remove them from the manuscript.

We have experimentally validated the interactions predicted with the two proteins studied in this paper: Ezh2 and Suz12. We agree with the Reviewer that the other interactors are only predictors. However, we totally disagree in considering these data as “misleading”. Several papers have been published in high-ranking journals so far that are solely based on catRAPID predictions. We believe that removing these data from the manuscript would be a missed opportunity for the community of scientists studying L1 RNAs (Long interspersed nuclear elements).

4. L332: “These data suggest that L1 RNAs may exert ‘sponge’ or ‘guide’-like mechanisms in regulating PRC2 recruitment to target sites and deposition of repressive epigenetic marks.” This statement is the closest thing for a model that was put forward in this manuscript, but it is a rather vague and yet a long stretch away from the data. The authors should set a hypothesis (or hypotheses) regarding a working model to explain how L1 regulates PRC2 and then set up to test it experimentally. To support their hypothesis, the authors would have to develop a significant line of research that would extend far beyond the preliminary data in Fig 3-4. First, they will have to experimentally identify how PRC2 and L1 RNA interact. This would require answering the question: what bases in L1 RNA are sufficient and required for binding to PRC2? Based on this investigation, the authors should be able to introduce point mutations to L1, that would disrupt its interactions with PRC2.

If it is practical to introduce these mutations to selected L1 elements in cells then it could be used in conjugation with ChIP-seq and RNA-seq to show that the recruitment and activity of PRC2 at the mutated locus has changed according to the hypothesis. Do L1 RNA regulate the activity of PRC2 at their site of transcription (i.e. in cis) or other loci (i.e. in trans)? This is a question that should be considered while setting up a model and designing experiments to test it. If the authors believe that the effect is rather global (e.g. global ‘sponge’) then they should identify the L1 region that is sufficient for binding to PRC2 and they should overexpress it with the expectation that it could rescue the PRC2-linked global effects seen after by L1 knockdown. The same experiments can then be carried out in the presence or absence of PRC2 inhibitor or knockout, with the expected result of epistasis in case the transcriptional changes are attributed to the hypothesized L1—>PRC2 pathway.

We thank the Reviewer for these important observations. Some of the suggested experiments are currently carried out in our laboratory and part of the next paper. Of note, it is very difficult to introduce mutations in single L1 genes. First, one needs to uniquely identify the several hundreds of single expressed L1s and carefully choose the representative ones to study, as correctly pointed out by the Reviewer. We are currently carrying out this type of analysis by taking advantage of Oxford Nanopore RNA direct sequencing.

5. Where in the nucleus the siRNA knockdown of L1 takes place? Is this type of perturbation affect the abundance of L1 RNA that is associated with facultative heterochromatin or is it take place in the nucleoplasm? Or co-transcriptionally? This question seems relevant and needs to be addressed experimentally as if the knockdown does not take place at the site of transcription it would exclude regulation in cis.

We thank the Reviewer for pointing out this very important and interesting question. Unfortunately, we consider it out of reach for this study.

Minor points

6. L49: The text says that “Ezh2 is the catalytic enzyme”, but PRC2 is the enzyme while EZH2 is the catalytic subunit.

7. Extended Data Fig.3: it is impossible to read the text within ED Fig 3b, which makes all the TSNE representations across ED Fig 3e-h obsolete.

Reviewer #4:

Summary of the article.

Previously Percharde et al. has shown that the transposon LINE1 (L1) expression is essential for embryonic stem cell self renewal in mouse, by acting as a nuclear scaffold RNA that activates ribosomal RNA (rRNA) synthesis and inhibits Dux, the master activator of a transcriptional program specific to the 2-cell embryo (PMID: 29937225). In addition to its expression in the pre-implantation embryos, L1 is also expressed in neuro-progenitor cells (NPCs) and the active copies can lead to somatic retrotranspositions and generate novel mobile element insertion (MEI) mutations. In this manuscript, Mangoni et al. explored the regulatory roles of L1 RNA in the neuro-progenitor cells, and discovered that:

1) L1 RNA can serve as regulatory non-coding RNA that contribute to the (A) differentiation of NPCs and (B) migration of the post-mitotic neurons.

2) The regulatory role of L1 RNA is through interacting with the subunits of the protein complex PRC2, and regulating its targets for histone methylation (H3K27me3), a marker for silenced chromatin.

The insight of L1 RNA serving as regulatory components, affecting the chromatin structure and contributing to the NPC differentiation is very exciting, however, a few key aspects of the supporting evidence need to be further clarified to strengthen the biological findings.

We thank the Referee for considering our work very exciting.

Major critiques

1) Systematic bias from the RNA interfering probes -- shL1-a and shL1-b
While the efficacy of shL1-a and shL1-b to knock down L1 expression has been shown in Fig. 1 and Fig. 2, their possible off-target effect needs to be evaluated with more details. L1 sequences are abundant in the mouse genome and many of them have been exonized into coding sequences, for instance, it was estimated that 100 – 300 L1 elements overlap with genes in the mouse genome (PMID: 17594509). Thus, it is very possible that the interfering RNA probes could knock down mRNAs of certain genes. This effect can be easily evaluated by looking for L1 fragments, particularly the shL1-a or shL1-b sequences, among the coding sequences of the top down-regulated genes.

We thank very much the Reviewer for his/her observation and giving us the chance to better clarify this point.

To this end, we searched for possible off-targets of shL1-a and shL1-b in down-regulated genes, in the *in vivo* and *in vitro* RNA-seq datasets. We performed the analysis allowing 0 or 1 mismatches for shL1-a and shL1-b sequences on the expressed transcriptome. No predicted off-targets with no mismatches have been found for shL1a while only one gene (*Fech*, ferrochelatase) is down-regulated by shL1-b *in vitro* and contains the shL1-b sequence with 0 mismatches.

When performing the analysis allowing 1 mismatch we found, again, only one gene (*Kcnq1ot1*, KCNQ1 overlapping transcript 1) which is down-regulated by both shL1-a and shL1-b *in vitro* and by shL1-a *in vivo* (log2FC: -1.34 and -0.55, *in vivo* and *in vitro* respectively). As a note of caution, we quoted in the paper that *Kcnq1ot1* seems to regulate chromatin status by recruiting chromatin-modifying complexes. Nevertheless, according to these results, the interfering approach with shL1-a/-b seems to be specific for L1 RNAs and appears not to have a relevant impact upon the expression levels of possible off-target genes which are down-regulated as a consequence of L1 RNAs knockdown.

We added these new important results to the revised version of the manuscript:

“Among genes down-regulated by shL1s, no predicted off-targets with no mismatches have been found for shL1a while *Fech* (ferrochelatase) is the only predicted off-target with no mismatches for shL1-b. *Kcnq1ot1* (KCNQ1 overlapping transcript 1) is the only predicted off-target with 1 mismatch with both shRNAs (Extended Data Table 8). As a note of caution, *Kcnq1ot1* is an antisense lncRNA recently shown to regulate chromatin status by recruiting chromatin-modifying complexes.”

In addition, the authors used two small RNAs, shL1-a and shL1-b, that target the similar regions in L1 as probes for RNA interference the in vitro experiment. While both shL1-a and shL1-b could downregulate L1 RNAs across different subfamilies and led to a large overlap of DEGs, it is still surprising to observe that across the whole transcriptome the shL1-b experiments still clustered much closer to the controls rather than the shL1-a experiments (Extended figure 3c). While this difference could be explained by truncated L1 RNAs that could only be interfered with either shL1-a or shL1-b, it remains possible the bias could be contributed by the difference in the off-targets.

Finally, despite the difference between shL1-a and shL1-b in the in vitro experiment, the authors only applied shL1-a in the in vivo experiment, and observed a significant number of genes that were regulated in opposite patterns from the in vitro experiment (line 191). To rule out the systematic bias introduced by the selection of the interfering probe, it is therefore essential to replicate the in vivo experiment with the shL1-b probe and reevaluate the comparison with the in vitro experiments.

We thank the Reviewer for highlighting this point allowing us to improve the manuscript.

As demonstrated by the off-targets analysis performed on *in vivo* and *in vitro* RNA-seq datasets, shL1-a and shL1-b seem to act specifically towards L1 RNAs, with only two genes as possible off-targets of shL1s.

As shown in Fig. 2c and in Extended Data Fig. 3a-b, in all the experiments done to achieve L1 RNAs knockdown, the efficiency of the silencing is higher for shL1-a than shL1-b. Although the two shRNA sequences have been designed to target the more conserved region of L1s (Orf2) and targeted sequences are quite close each other, our results indicate that shL1-a is the one with the better ability to target L1 transcripts. Since the off-target analysis rules out the possibility that shL1-a and shL1-b could have distinct effects on the basis of a different ability to act upon off-target genes, our opinion is that differences in changes of gene expression promoted by shL1-a and shL1-b are possibly due to the lower efficiency of L1 RNAs knockdown promoted by the latter.

To assess the effect of shL1-b *in vivo*, we delivered shL1-b to the mouse brain neocortex by *in utero* electroporation at E12.5 and looked at major effects promoted by shL1-a, for instance the ability to reduce the progenitor cells pool, and to induce neuronal commitment.

As showed in Extended Data Fig. 1b-c, shL1-b recapitulated the same effects of shL1-a at E14.5 by significantly decreasing the number of Pax6⁺ cells (progenitor cells) and increasing NeuroD1⁺ cells (neuronal committed cells). ShL1-b has been already used in the following work (PMID: 29937225).

2) The function of L1 RNA through interacting with PRC2 and modifying histone methylation
In the second half of the manuscript, the authors provided supporting evidence for a model where L1 RNAs interact with protein Suz12 and Ezh2 and regulate the histone methylation activities (H3K27me3) of the PRC2 complex. They showed that the localization of L1 RNAs are significantly associated with chromatin, and a significant amount of L1 RNAs can be co-IPed with both Suz12 and Ezh2, when using a house keeping gene (GAPDH) and 45s rRNA as controls. Since Suz12 and Ezh2 are primarily localized in the chromatin, this co-IP enrichment of L1 RNAs could be due to the localization of the RNAs, instead of direct interactions with the suggested proteins. It is therefore essential to replicate the co-IP results using the chromatin fraction of RNAs (instead of the whole cell lysate), or using other control RNAs with similar localization preference but has no interactions with the PRC2 complex.

We thank the Reviewer for highlighting this point allowing us to improve the manuscript.

As suggested to avoid co-immunoprecipitation due to the bias of the localization, we performed the RNA IP on the chromatin RNA fraction and we used U1 snoRNA as additional control for chromatin-associated RNA and negative control for PRC2 binding (Kanhare A, Viiri K, Araújo CC, Rasaiyaah J, Bouwman RD, Whyte WA, Pereira CF, Brookes E, Walker K, Bell GW, Pombo A, Fisher AG, Young RA, Jenner RG. Short RNAs are transcribed from repressed polycomb target genes and interact with polycomb repressive complex-2. Mol Cell. 2010 Jun 11;38(5):675-88. doi: 10.1016/j.molcel.2010.03.019. PMID: 20542000; PMCID: PMC2886029). Also in this case, we observed enrichment for L1 RNAs in the IP compared to IgG but not for 45s rRNA, Gapdh and U1, as expected.

The authors then investigated the H3K27me3 statuses of the DEGs in the L1-knock down experiments, and showed that in a panel of 16 (8 upregulated and 8 downregulated) genes, the changes of H3K27me3 and the Ezh2 binding were consistent with the changes of gene expression. While this is a very strong evidence supporting the proposed mechanism of the L1 RNA, 16 genes remain a small subset of all DEGs. It will be important to demonstrate how representative these 16 genes are and if the pattern can be extrapolated towards other genes. Specifically, how did these genes were selected in the first place, and what about the changes in other DEGs? More robust evidence can be provided with ChIP-seq analyses to pull down Ezh2 and H3K27me3 marks in the same L1-knock down and control cell cultures, just like their ChIP-qPCR experiment, but to investigate all genes instead of a limited gene panel.

We thank the Reviewer for finding our results a very strong evidence and for his/her suggestions. In the revised version of the paper, we have also performed ChIP-seq experiments to evaluate both deposition of H3K27me3 and binding of Ezh2, in a genome-wide scale. First, we observed a significantly higher H3K27 three-methylation upon regulatory regions of down-regulated genes, accordingly with RNA-seq experiments, suggesting that down-regulation of a significant portion of genes after L1 RNAs silencing is due to hypermethylation of their promoter region. No significant changes were observed for genes up-regulated or whose expression didn't change according to RNA-seq. Ezh2 binding was altered by L1 RNAs silencing for 418 genes.

The majority of them, 261 (62%; $P = 7.6E-11$), overlapped with H3K27me3 hypermethylated regions (Fig. 4c), suggesting that L1 RNAs knockdown was able to alter, at least for these regions, both Ezh2 occupancy and H3K27me3 deposition. This group of genes are enriched for transcription factors involved in neuronal differentiation. Future work will address the details and functional outputs of PRC2-L1 RNAs interactions at single gene loci and the significance of the difference in the magnitude of variations between H3K27me3 deposition and Ezh2 binding upon L1 RNAs silencing. Of note, the previous Ab used for ChIP of Ezh2 did not pass the standard of quality controls when used in ChIP-seq experiments. Therefore, in this revised version we decided to eliminate previous data on ChIP experiments on single gene target. These new experiments are described in the revised version of the manuscript in the paragraph: **L1 silencing impairs cortical cell development and maturation *in vitro***.

Minor critiques

1) It is essential to show the exact p values (e.g., $p=1.1e-07$, line 170), instead of the range of the p values (e.g., $p < 0.01$, line 69). In the *in vivo* experiments (line 68-81), it is also unclear how many different comparison tests were performed, and whether the p values were corrected for the number of comparisons (e.g., multiple testing correction).

We thank the Reviewer for this suggestion.

In the revised version of the manuscript, we showed the exact p values instead of the range of p values. Unfortunately, this was not possible for the range $P < 0.0001$ when statistical analysis was performed with GraphPad since the software does not show the exact p value for that interval. So, in this case we kept the previous annotation ($P < 0.0001$). The numerosity of the experimental groups in each statistical analysis is reported in figure legends as recommended by editorial guidelines, as well as the post-test used (Unpaired t test with Welch's correction).

2) The ChIP-qPCR experiment (Figure 4) lacks proper negative controls such as genes with no expression changes in the L1-knock down culture.

This figure has been eliminated since we carried out a Chip-seq experiment.

REVIEWER COMMENTS

Reviewer #1 (Remarks to the Author):

I have carefully read the answers of the authors and looked at their experimental and editorial modifications.

I am fully satisfied by the answers

Reviewer #2 (Remarks to the Author):

The authors did not address any of my major points, which makes me wonder what is the point of including me within this peer review process. Key statements in this manuscript are still unsupported, including the statement in the title that LINE-1 RNAs “tuning PRC2 activity”: this could be either right or wrong given the experiments that were carried out. I reiterate my original recommendation for rejection.

Information regarding the major points that were not addressed:

Point 1: The authors were asked to moderate an unsupported statement in L200 (“In summary, L1 RNAs regulate chromatin remodelling and transcriptional control”). Instead, they focus their response on the significance, which I leave to the editor. Yet, the authors still did not moderate that statement, which is yet unsupported, given that global effects were not considered and causality was not established (see point 4 below).

Point 2: The authors did not indicate how they discussed the limitation of the knockout textually, although that was all they were asked to do.

Point 3: The authors did not address this point experimentally and are still substantiating their statements based on predictions.

Point 4: This point was not addressed experimentally, which was required in order to establish causality between L1 to PRC2 activity. Without establishing causality, key

statements in this manuscript are not supported, including the title itself.

Point 5: The authors choose not to address this point.

Reviewer #4 (Remarks to the Author):

The authors have made substantial revisions to the manuscript, and I appreciate that they have added several new experiments and technical controls. However, after carefully reviewing the new results, I still have concerns about the overall consistency of the molecular signatures from LINE1 knockdown experiments. For instance, the authors identified:

1) In the in vivo knockdown:

- a. the upregulated genes are enriched with “nervous system development, axogenesis, and projection development”;
- b. the downregulated genes are enriched with “protein synthesis, mitochondrial activity, ribosome assembly, and cell division”.

2) In the in vitro knockdown experiments:

- a. the upregulated genes are enriched in “RNA processing, splicing, transcription, chromatin remodeling, and DNA binding”;
- b. the downregulated genes are: “synaptic transmission, ion transport, channel activity and neurotransmitter receptor activity”.

3) In the transposon RT inhibitor experiment:

- a. the up-regulated genes are enriched for terms related to “inhibitory synapse assembly, glycine transport, and behavior”
- b. the down-regulated genes are enriched for “cell cycle and mitosis”

4) In the L1knockdown followed by H3K27Me3 or Ezh2 ChIP-seq experiments:

- a. Upregulated genes (irrelevant)
- b. The down-regulated and hypermethylated genes (in H3K27Me3) are enriched with

“neurogenesis, synaptic transmission, neuronal differentiation, regulation of Wnt signalling pathway, and brain developmental processes”. This is particularly confusing, since the authors suggest L1 helps to limit neuronal commitment (line 89), so L1-KO should lead to upregulation of the neurogenesis genes.

How do these findings compare with each other? It may be possible to improve the representation of the GO term analysis with a GO term enrichment map. We appreciate the authors’ attempt to explain some of the differences in the discussion. However, without a clear and consistent message from these experiments, the manuscript remains confusing to read and unfortunately the results felt superficial.

Here are some additional comments, mostly centered around the consistency across experiments:

- 1) Extended figure 1i was cited in the wrong place in the manuscript.
- 2) Line 111: “indicated 1129 common genes, of which 215 ($P = 7.71e-33$)”. What is the significance of the overlap of 1129 common genes? Were they consistently upregulated and downregulated in both experiments? It’s also unclear what the P value is testing on.
- 3) Line 143: “3028 common DEGs”. What is the statistical significance of the overlap?
- 4) Line 191: “10.4% of genes altered by shL1-a and -b”. What are the directions (i.e., upregulation and downregulation) and the statistical significance of the overlap?
- 5) Line 303: “FDR < 0.05”. Please add the 95% confidence intervals to Figure 4a to support the statistical significance. If L1-gene regulation is partially through Ezh2 and the histone modification, what is the correlation between gene down-regulation and H3K27Me3 hypermethylation or Ezh2 binding? The consistency between H3K27Me3 hypermethylation and Ezh2 binding as shown in Figure 4b is less significant as that's the known function of Ezh2. Finally, the changes in Ezh2 binding in Figure 4d are hardly discernible.
- 6) Were there no biological replicates in the CHIP-seq analysis? If so, how did the statistics calculated in Figure 4b?

Reviewer's Comments:

Reviewer #2 (Remarks to the Author)

The authors addressed most of the points that I raised or provided acceptable explanations for most experiments they wish to leave for future studies. I believe that this manuscript will be suitable for publication after the authors will address the following two minor points:

1. The authors still did not demonstrate causality between L1-binding by PRC2 to the regulation of PRC2 (point 4 in my previous reviews). Nevertheless, the authors provided an acceptable explanation for why establishing a mechanism (and causality) would require further development beyond this study and they also moderated some related statements in the main text. Yet, that means that the following statement within the abstract is still unsupported: "L1 RNAs exert some of their regulatory functions by interacting with the Polycomb Repressive Complex 2" (abstract, L24-25). Specifically, the data in the manuscript indicate that PRC2 binds to L1 and, separately, that L1 depletion leads to changes in the ability of PRC2 to bind and modify some target genes, but the authors did not show that L1 exert regulatory functions "by interacting" with PRC2. The authors did well in moderating related unsupported statements within the text but it would be appropriate if they will do so also in the abstract.

We thank the Reviewer for her/his comment. We amended the abstract accordingly:

"In cortical cultured neurons, L1 RNAs are mainly associated to chromatin and interact with the Polycomb Repressive Complex 2 (PRC2) protein subunits *enhancer of Zeste homolog 2* (Ezh2) and *suppressor of zeste 12* (Suz12). L1 RNA silencing influences PRC2's ability to bind a portion of its targets and the deposition of tri-methylated histone H3 (H3K27me3) marks."

2. Fig S6 was modified, but the resolution of some of the panels is too low for reading axes legends (mainly S6g,h).

We thank the Reviewer for her/his comment. We modified the Extended Data Fig. 6 making panels g and h 50% larger. Also, we increased the text size of some other elements (i.e. axis legends and labeling). These changes should help the readability of the figure.

Reviewer #4 (Remarks to the Author)

I have considered in detail the new changes made by the authors to clarify my previous concerns in regard to the consistency of the measured impacts from L1 KO. There is now sufficient evidence for (1) L1 RNA plays a role in neurogenesis and gliogenesis and (2) likely functions partially through interactions with PRC2 and Suz12. However, I am still concerned that the specific roles of L1 RNA in this process have yet to be fully characterized, likely due to the limitations of the high throughput omic analyses.

Here is another summary of the major experiments to quantify the L1 KO impacts:

- 1) In vivo KO;
- 2) In vitro KO;
- 3) RT inhibition;
- 4) H3K27Me3 ChipSeq;
- 5) Ezh2 ChIPseq

In this new revision, the authors explained well about the difference among the (1) in vivo, (2) in vitro, and (3) RT inhibition analyses, likely due to the developmental timing or the specific regulation mechanism (through RNA, not RT). The authors also showed significant consistency among the biological replicates and the direction of the gene regulations (e.g., higher H3K27Me3, more Ezh2 binding, and less transcription).

Mechanistically, however, there still doesn't seem to be any clear developmental pathways that are consistently regulated as measured across most of these five assays.

A. To compare between 2) in vitro and 4) methylation, the authors said: "Consistent with GO terms enriched by down-regulated genes after L1 RNAs silencing (Extended Data Table 3, Fig. 2 f, bottom), GO analysis of the fraction of these genes contained in hypermethylated regions revealed an enrichment for biological processes related to regulation of neuronal development and for functionalities that include calcium ion concentration, action potential, neuron projection development, synaptic transmission, assembly, plasticity and potentiation (Extended Data Table 7, Fig. 3g)" [[row 274-279]].

However, I failed to see the "regulation of neuronal development" highlighted in the word cloud (Fig. 2f). It only shows up as the 83th most enriched GO category in the extended data Table 3. As a comparison, the "positive regulation of neuronal development" showed up as the 11th most enriched category in the methylation analysis. I am unsure if the (1) vastly different significance and (2) different GO terms can warrant such a strong statement about the consistency of the results.

We thank the Reviewer for her/his comment that give us the opportunity to improve our article by avoiding too strong statements. We agree that the two lists of GO terms (enriched for all the genes down-regulated by L1 silencing and for those that are down-regulated and contained in peaks/islands

with a significantly higher deposition of H3K27me3) do not perfectly match and show some differences. We therefore reasoned that in our experiments H3K27 hypermethylation targets downregulated genes that are important for processes related to neuronal activity and functionalities supporting the notion that hypermethylation of H3K27 could represent **one, but not the only one**, of the possible mechanisms by which L1 RNAs knockdown could affect gene expression changes and consequent neuronal maturation and functions.

With this reasoning in mind and to meet the Reviewer's comments, we amended the related text accordingly (rows 239-245):

“GO analysis of the fraction of down-regulated genes contained in hypermethylated regions revealed an enrichment for biological processes related to neuronal cells' functions and activities including calcium ion concentration, action potential, cell junctions and membrane rafts assembly, neuron projection development, synaptic transmission, assembly, plasticity and potentiation (Extended Data Table 7, Fig. 3g).

These results suggest that changes in repressive epigenetic marks may account for a portion of the effects on gene expression observed after L1 RNAs silencing in cultured cortical cells.”

B. To compare between 2) in vitro, 4) methylation, and 5) Ezh2 binding, the authors said: “Interestingly, genes affected by both increased deposition of Ezh2 and H3K27me3 mark were important for transcriptional regulation of neuronal fate commitment and glial and oligodendrocyte lineages specification as well as for neuronal cell functionalities, including development of neuronal projections and synaptic processes (Extended Data Table 7, Fig. 4g).” [[row 335 -341]].

And

“These data suggest that L1 RNAs may influence PRC2-dependent deposition of repressive epigenetic marks on genes crucial for neuronal cell differentiation and activity, a prediction that has been validated with ChIP-seq experiments.” [[row 391-394]].

May I ask why the authors altered the name of the GO category “positive regulation of nervous system development” (as shown in Extended Table 7) to “nervous system development” in the main Figure 3g? In addition, the authors altered the name of the GO category “positive regulation of neuron differentiation” (as shown in Extended Table 7) to “neuron differentiation” in the main Figure 4g. I doubt it is due to the limited space, as the authors used the proper labeling in the first manuscript but changed it in the recent revision.

We thank the Reviewer for her/his comment pointing out these inconsistencies. While these modifications were exclusively due to formatting needs, we realize they can generate unwanted confusion in the readers. We apologize for this misunderstanding and amend Fig.3 and Fig.4 accordingly to the Reviewer's suggestion to show the full name of GO categories.

The critical contradiction here, is that the authors suggested the L1 RNAs play a role in limiting neuronal commitment (row 85, the in vivo assay) and L1 KO promotes neuron differentiation, but in

the methylation analysis and Ezh2 binding assays, the authors showed the genes that are positively regulating neuron differentiation are significantly “down-regulated” after L1 KO. By the way, here is the definition of this GO category from AmiGO -- any process that activates or increases the frequency, rate, or extent of neuron differentiation.

I highlighted this issue in my previous review, and I am disappointed to see it remain unanswered and instead changed towards what seems to be a cover-up in the main figures. Despite the difference between the *in vitro* and the *in vivo* experiments and the fact that several genes play a role in both positive and negative regulations, the authors should still discuss the seemingly opposite impacts of L1 KO for genes involving neuron differentiation.

We thank the Reviewer for her/his comment. We apologize if the discussion of this part of the work turned out to be ambiguous and contributed to arise the point that there are contradictory results from different experiments and/or development stages. It's our fault for not properly explain the meaning of our findings.

As properly pointed out by the Reviewer, it is a fact that the two model systems present crucial differences that we discussed in the second revised version of the manuscript at the end of the Paragraph: L1 silencing impairs cortical cell development and maturation *in vitro* (previous rows 187-197). We thank the Reviewer for considering this text appropriate.

Nevertheless, we now realize that we have not adequately discussed a crucial element of L1s biology, their heterogeneity which is probably at the basis of the issue raised by the Reviewer. According to L1Base, the mouse genome contains about 2800 full length intact L1 from different subfamilies and therefore a cell may express hundreds of different independent full length L1 transcripts (<https://doi.org/10.1093/nar/gkw925>, <https://doi.org/10.1101/gr.198301>, <https://doi.org/10.1186/s13100-022-00269-z>, <https://doi.org/10.2741/e537>). Their biological activity can be determined by their location in the genome (i. e. intergenic, intronic, promoter or enhancer-associated) and their length (full length, 5' and 3' truncated). Furthermore, their expression is differentially regulated according to the TFBS content of their promoters. By taking advantage of targeting a large repertory of heterogeneous L1 RNAs, we miss the complexity of the regulated L1 transcriptome and the function associated to single transcripts. Importantly, the cellular repertory of L1s can be different in neuronal progenitor cells and in differentiated neurons (<https://doi.org/10.1101/gr.198301>, <https://doi.org/10.1186/s13100-022-00269-z>, <https://doi.org/10.2741/e537>) while being commonly targeted by ShL1 silencing for their sequence homology.

Therefore, we decided to remove any strong conclusion and move comments on the possible causes about the differences between the two model systems from the Result section (previous rows 187-197) to the Discussion and add a commentary on the heterogeneity of L1s, concluding with the considerations that further studies are needed in order to solve such a complicated issue.

(rows 323-340):

The following text is now present in the Discussion:

“It remains to be determined how L1s can promote both neuronal progenitor proliferation *in vivo* and neuronal differentiation *in vitro*. This is mirrored by the opposite pattern of expression of genes that were upregulated *in vivo* and downregulated *in vitro* and involved in synaptic transmission, signalling, ion transport and neural activity (Extended Data Table 6, Extended Data Fig. 5a). This behaviour could be the consequence of the time of harvesting of cortical neurons for *in vitro* differentiation (E17.5), modelling later stages of the neurodevelopmental cascade with respect to the E12.5 stage tested *in vivo*. Furthermore, the time required for proper AAV-mediated expression of shL1s in cultured cells may delay further the consequences on neuronal maturation and cell type composition. Most of the pro-neurogenic effect *in vivo* could be due to the impact of L1 silencing upon proliferation and commitment of progenitor cells, which are much less represented at E17.5. Importantly, the difference in the effect of L1 silencing may be caused by the heterogeneity of L1 transcripts. Their biological activity may depend on the location in the genome (i. e. intergenic, intronic, promoter or enhancer-associated), length (full length, 5’ and 3’ truncated), protein interaction network and expression, being differentially regulated according to the TFBS content of their promoters. By taking advantage of targeting a large repertory of heterogeneous L1 RNAs, the complexity of the regulated L1 transcriptome and the function associated to single transcripts, are missed. Unveiling the repertory of single L1 transcripts by third generation sequencing technologies, will be instrumental to better define their specific functions in the two model systems.”

Additionally, the authors mentioned adding shaded areas in Figure 3e to represent the confidence intervals, but there are no such areas in the new figure -- it looks exactly like the old one.

We thank the Reviewer for her/his comment. We apologize for the misunderstanding about this point. As we mentioned in the previous response letter, and in the relative figure legend, we amended Fig. 3e adding the 95% confidence intervals as shaded areas. However, variations between replicate samples in the ChIP-seq experiment are so small that is very hard to appreciate the shaded areas in the figure by eye. The Reviewer may notice that they become visible by magnifying the figure.

I apologize for further dragging on the reviewing process. However, the exact impacts of L1 RNA in neuron/glia differentiation remain unclear in the current state of the manuscript. I also suggest the authors use full GO category names in the figures, to avoid unnecessary confusion in interpreting the results.

We are deeply grateful to the Reviewer for her/his time and effort to improve our manuscript. We respectfully think that in this new revised of the manuscript we have addressed all her/his points.

Reviewer #2

The authors did not address any of my major points, which makes me wonder what is the point of including me within this peer review process. Key statements in this manuscript are still unsupported, including the statement in the title that LINE-1 RNAs “tuning PRC2 activity”: this could be either right or wrong given the experiments that were carried out. I reiterate my original recommendation for rejection.

We thank the Reviewer for her/his comment. We confess that in our previous version we have been confused by some of the comments received. Nevertheless, in this new revised version we have made a major effort to address the large majority of her/his points.

Firstly, we have changed the title of the paper from:

“LINE-1 RNAs regulate cortical development by tuning PRC2 activity”

to

“LINE-1 regulates cortical development by acting as long non-coding RNAs”

since the previous version may imply that the majority of L1-dependent phenotypes were due to the binding to PRC2. We agree that L1 RNA-PRC2 interaction is one, although important, of the many mechanisms by which L1 RNAs can regulate corticogenesis.

To better formulate this distinction, we have also moved ChIP-seq data on H3K27me3 to the new paragraph “L1 RNAs are mainly associated to chromatin and influence the pattern of H3K27me3 deposition” while we have maintained ChIP-seq data on Ezh2 in the last paragraph of the Results section, now intitled “L1 silencing influences Ezh2 activity to target genes in cortical cultured neurons”.

Furthermore, we have added to the Discussion section, the following texts:

Lines 333-344:

“Nevertheless, the outcomes of L1 RNAs silencing seem to be mainly associated to activities as regulatory lncRNAs and in particular to their binding to chromatin. According to a model of action, L1 RNAs may cause phenotypic changes through their direct or indirect impacts on chromatin remodelling complexes and transcription factors activities, as found for lncRNAs (18). Binding sites for pro-neurogenic transcription factors were indeed enriched in regulatory regions of genes differentially expressed *in vivo* and *in vitro* upon L1 RNAs silencing. A large number of TSS of genes down-regulated according to RNA-seq experiments, were hypermethylated at H3K27me3, a repressive epigenetic mark. To understand how L1 RNAs may influence these nuclear activities, protein interactors were predicted computationally and validated *in silico* for the presence of experimental eCLIP data. They provide candidate regulatory networks to be further studied for the functional outputs of L1 RNAs expression during cortical development.”

Lines 379-382 (last sentence of the Discussion):

“Since G4s operate as common binding hubs for many transcription factors to promote increased transcription of G4-containing genes (32, 33, 34), we speculate that L1 RNAs can influence the activity of other transcriptional networks by similar mechanisms and that these interactions have been relevant in the evolution of the brain.”

We have also addressed the remaining observations/critiques as follows.

Information regarding the major points that were not addressed:

Point 1: The authors were asked to moderate an unsupported statement in L200 (“In summary, L1 RNAs regulate chromatin remodelling and transcriptional control”). Instead, they focus their response on the significance, which I leave to the editor. Yet, the authors still did not moderate that statement, which is yet unsupported, given that global effects were not considered and causality was not established (see point 4 below).

We thank the Reviewer for her/his comment. To moderate our statement in L200, we have modified the text as follows:

“In summary, L1 RNAs are continuously required for proper corticogenesis, but with different effects on the same group of genes, according to the timing of differentiation of neural progenitors and maturation of single cell types.”

Point 2: The authors did not indicate how they discussed the limitation of the knockout textually, although that was all they were asked to do.

(Previous comment: Fig 1b and 2c: The data in Figures 1b and 2c demonstrate a 2-fold reduction in the L1 expression level, at best. This is a key limitation of the experimental system. If knockdown efficiency cannot be improved then the authors should at least discuss this limitation textually.)

We thank the Reviewer for the observation and the possibility it gave us to better clarify this point in the text of the manuscript. As reported, LINE-1 RNAs levels after knockdown with both the two shRNAs were 50-60% lower than control conditions, both in *in vivo* and *in vitro* experiments. shL1-a sequence was chosen since it showed the highest knockdown efficiency among about 20 sequences tested (*data not shown*). shL1-b was previously used in another study (doi: 10.1016/j.cell.2018.05.043). This level of efficiency of the knockdown of L1 expression can be compared to that produced by different methods, and with different targeting sequences (see in doi:10.1038/ng.3945 and in doi: 10.1016/j.cell.2018.05.043). The major limitations to obtain a higher silencing efficiency can be due to some specific features of L1 sequences. Firstly, L1s transcripts are among the most abundant transcripts in a cell and mostly located in the nucleus and bound to chromatin (>90%, as reported by our experiments); indeed, formation of secondary structures and interaction with proteins, both at ribonucleoprotein particles and chromatin, contribute to making L1

RNA extremely difficult to be accessed with target sequences that are possibly masked by these interactions. Most importantly, despite the high degree of conservation, the diversity in terms of sequence of the actively transcribed L1 elements in a mouse cell gives rise to very heterogeneous populations of RNAs. In order to target the majority of the transcribed L1s, we have taken advantage of shRNAs designed on the most conserved region of L1 sequences (Orf2 CDS); however, a number of L1 transcripts could be naturally resistant to these shRNAs and escape silencing. All of these points make L1 RNAs difficult to be targeted with RNA interference approaches.

We amended this point in the text accordingly (lines 110-114):

“RT-qPCR confirmed the reduction of L1MdA, L1MdGf, and L1MdTf subfamilies and on the conserved sequence of L1 Orf2. shL1-a showed a L1 knockdown efficiency of 40-45% and shL1-b of 35-40%, compared to non-infected (n.i.) cells or cells infected with a control shRNA (shCtrl; Fig. 2c). Many factors may negatively affect the efficiency of the knockdown exerted by shL1s including localization of L1 RNAs, formation of secondary structures, interactions with DNA and proteins that could mask the sequences targeted by the shRNAs, and the high heterogeneity of transcribed L1 elements which gives rise to different populations of L1 transcripts in terms of RNA sequences, some of them resistant even to shRNA designed on highly conserved regions”.

Point 3: The authors did not address this point experimentally and are still substantiating their statements based on predictions.

(Previous comment: Fig 3f end ED 5e-f: The catRAPID analysis is merely predictive and misleading without experimental validation. The authors should either validate these predictions experimentally or remove them from the manuscript.)

We thank the Reviewer for giving us the chance to better clarify this point. Data regarding the binding motifs of Suz12 is not just predictive, since catRAPID v2 software integrates a database containing high-quality motifs for a list of RNA-binding proteins that were previously validated by eCLIP, therefore experimentally. Revised Fig. 4a-b shows the number of L1 RNAs sequences harboring at least one Suz12 binding motif (Fig. 4a) and the number of Suz12 binding motifs harbored by different L1 families (Fig. 4b), respectively. That's not a prediction, but the count of how many times a certain sequence harbors a certain motif, and of how many times a certain sequence appears inside an L1 element.

We respectfully note that the fact that catRAPID predictions and the motifs analysis agree on the binding of Suz12 is not trivial, as previously shown for the long non-coding *Xist*, where the software predicts that the protein does not bind directly (<https://pubmed.ncbi.nlm.nih.gov/23093590/>), as subsequently confirmed by experimental work (<https://pubmed.ncbi.nlm.nih.gov/32898472/> and <https://pubmed.ncbi.nlm.nih.gov/31061525/>).

Accordingly, we have modified the text as follows: (lines 259-262):

“Combining secondary structure, hydrogen bonding, van der Waals contributions and experimental enhanced crosslinking and immunoprecipitation (eCLIP) data, *catRAPID* estimates the binding propensity of protein-RNA pairs.”

Finally, Extended Data Fig. 6d-f and Fig. 4d-e experimentally demonstrate that Suz12 binds L1 RNAs *in vivo* by RNA immunoprecipitation, both on total and chromatin-enriched lysates, respectively. From our point of view, the message of this part of the work is not to characterize at a deep structural level the interaction sites on RNA sequences bound by PRC2 proteins Ezh2 and Suz12, but to demonstrate that an interaction between the two PRC2 proteins and L1 RNAs does occur in mouse neurons.

In another project in the lab, we have confirmed that L1 RNAs are bound by FMRP in neurons, validating another element of the list. Of note, we believe that “Table 9. Ranking *catRAPID* score RBPs” suggests new candidate mechanisms for L1 RNAs regulatory functions of cortical development beyond the interaction with PRC2, strengthening the vision of the Reviewer (please see again lines 333-344 in the Discussion).

Point 4: This point was not addressed experimentally, which was required in order to establish causality between L1 to PRC2 activity. Without establishing causality, key statements in this manuscript are not supported, including the title itself.

(Previous comment: L332: “These data suggest that L1 RNAs may exert ‘sponge’ or ‘guide’-like mechanisms in regulating PRC2 recruitment to target sites and deposition of repressive epigenetic marks.” This statement is the closest thing for a model that was put forward in this manuscript, but it is a rather vague and yet a long stretch away from the data. The authors should set a hypothesis (or hypotheses) regarding a working model to explain how L1 regulates PRC2 and then set up to test it experimentally. To support their hypothesis, the authors would have to develop a significant line of research that would extend far beyond the preliminary data in Fig 3-4. First, they will have to experimentally identify how PRC2 and L1 RNA interact. This would require answering the question: what bases in L1 RNA are sufficient and required for binding to PRC2? Based on this investigation, the authors should be able to introduce point mutations to L1, that would disrupt its interactions with PRC2.

If it is practical to introduce these mutations to selected L1 elements in cells then it could be used in conjunction with ChIP-seq and RNA-seq to show that the recruitment and activity of PRC2 at the mutated locus has changed according to the hypothesis. Do L1 RNA regulate the activity of PRC2 at their site of transcription (i.e. in cis) or other loci (i.e. in trans)? This is a question that should be considered while setting up a model and designing experiments to test it. If the authors believe that the effect is rather global (e.g. global ‘sponge’) then they should identify the L1 region that is sufficient for binding to PRC2 and they should overexpress it with the expectation that it could rescue the PRC2-linked global effects seen after by L1 knockdown. The same experiments can then be carried out in the presence or absence of PRC2 inhibitor or knockout, with the expected result of epistasis in case the transcriptional changes are attributed to the hypothesized L1—>PRC2 pathway.)

We thank the reviewer for his/her comments that aim to better characterize the interaction between PRC2 and L1 RNAs and to better understand how this interaction is impacting the epigenetic regulation of gene expression.

L1 elements are present in thousands of genomic loci, and as a pool of hundreds of different sequences of transcribed RNAs in a mouse cell; this high complexity and heterogeneity in terms of number and differences of L1 sequences raises many questions when planning experiments with the over-expression of a single wild type or mutated sequence. When we choose one single L1 element for over-expression, we negate the importance of hundreds of different transcribed L1 RNAs. We respectfully think that this complexity severely limits the significance of the experiment. Furthermore, it is unclear whether a portion of a L1 sequence harboring the predicted binding site for a protein may emulate the behavior of a full-length L1 RNA element.

In our laboratory, we are carrying on preliminary experiments aimed to build a more accurate mechanistic model, such as the investigation of the “*cis*-” or “*trans* mechanisms”, and the minimal sequence of L1 elements required to bind Ezh2 and Suz12. However, long-term experiments are required to properly address these important questions.

Nevertheless, we agree with the reviewer that the mechanism of action is not fully characterized. Therefore, as described above, in this revised version of the paper we have been very cautious in our interpretation of the data as shown by the change in the manuscript title and by the new paragraph in the Discussion pointing out that L1 RNAs-PRC2 interaction is just one of the several mechanisms involved in the establishment of L1-dependent phenotypes in cortical development.

Point 5: The authors choose not to address this point.

(Previous comment: Where in the nucleus the siRNA knockdown of L1 takes place? Is this type of perturbation affect the abundance of L1 RNA that is associated with facultative heterochromatin or is it take place in the nucleoplasm? Or co-transcriptionally? This question seems relevant and needs to be addressed experimentally as if the knockdown does not take place at the site of transcription it would exclude regulation in cis.)

We thank the reviewer for raising this very interesting point. As suggested, we checked for the efficiency of L1 knockdown in different cell compartments: cytosol, nucleoplasm and chromatin. Since L1 RNAs knockdown gave as overall result of 50-55% efficiency and because >90% of L1 RNAs were chromatin-bound, we found as expected that shL1-a and -b were able to decrease L1 RNAs levels not only in the cytosol and nucleoplasm (~60-70% of the cytosolic or nucleoplasmic L1 levels, i.e. from 4-5% to 2% of the total L1 RNAs), but also at the chromatin level even if with a lower efficiency (~40-45% of the chromatinic L1 RNAs level, i.e. from 90% to 60-65% of the total L1 RNAs). The chromatinic decrease of L1 RNA levels produced by shRNAs represents the majority of L1 sequences affected by the knockdown. This data suggests that the effects induced by L1 RNAs silencing are mainly due to perturbation of their activity at the chromatin where, according to RNA immunoprecipitation experiments on fractionated cell lysates, the PRC2-L1 RNAs interaction takes place.

We included this result to Figure 3d, and we amended this point in the text accordingly (lines 230-233):

“To test the ability of shL1s to reduce the amount of chromatin bound L1 transcripts, we analysed the knockdown efficiency of shL1-a and -b in subcellular RNA fractions. Cytosolic and nucleoplasmic L1 RNAs were both decreased of 60-70%, while chromatinic L1 transcripts by 30-40% (Fig. 3d).”

Reviewer #4

The authors have made substantial revisions to the manuscript, and I appreciate that they have added several new experiments and technical controls. However, after carefully reviewing the new results, I still have concerns about the overall consistency of the molecular signatures from LINE1 knockdown experiments. For instance, the authors identified:

1) In the in vivo knockdown:

- a. the upregulated genes are enriched with “nervous system development, axogenesis, and projection development”;*
- b. the downregulated genes are enriched with “protein synthesis, mitochondrial activity, ribosome assembly, and cell division”.*

2) In the in vitro knockdown experiments:

- a. the upregulated genes are enriched in “RNA processing, splicing, transcription, chromatin remodeling, and DNA binding”;*
- b. the downregulated genes are: “synaptic transmission, ion transport, channel activity and neurotransmitter receptor activity”.*

3) In the transposon RT inhibitor experiment:

- a. the up-regulated genes are enriched for terms related to “inhibitory synapse assembly, glycine transport, and behavior”*
- b. the down-regulated genes are enriched for “cell cycle and mitosis”*

4) In the L1 knockdown followed by H3K27Me3 or Ezh2 ChIP-seq experiments:

- a. Upregulated genes (irrelevant)*
- b. The down-regulated and hypermethylated genes (in H3K27Me3) are enriched with “neurogenesis, synaptic transmission, neuronal differentiation, regulation of Wnt signalling pathway, and brain developmental processes”. This is particularly confusing, since the authors suggest L1 helps to limit neuronal commitment (line 89), so L1-KO should lead to upregulation of the neurogenesis genes.*

How do these findings compare with each other? It may be possible to improve the representation of the GO term analysis with a GO term enrichment map. We appreciate the authors’ attempt to explain some of the differences in the discussion. However, without a clear and consistent message from these experiments, the manuscript remains confusing to read and unfortunately the results felt superficial.

We are deeply indebted to the Reviewer for her/his comments since they allow us to better clarify these important points here and in the manuscript. As discussed throughout the paper, the use of the very same shRNA against L1s gave different, or even opposite, results when comparing *in vivo* and *in vitro* models of neuronal development. At this time, we can only speculate about the differences of the two biological systems: a. the different developmental stages (early cortical development mainly involving proliferation of progenitor cells and neuronal commitment vs later neuronal cell maturation and acquisition of synaptic activity); b. the presence of a large population of proliferating cells at E14.5 *in vivo*, while most of the cells in the *in vitro* cultures are post-mitotic; c. the heterogeneity of cell populations in the mouse cortex (comprising different types of progenitors, neuronal committed, migrating, and differentiated cells) vs differentiated and matured neuronal-enriched populations in

culture; d. the different RNA interfering approaches used for the two models (transfection by i.u.e. *in vivo* vs expression by AAV infection *in vitro* where shL1s are expressed under a promoter that is activated even later in the differentiation cascade); e. the different pools of L1 elements transcribed in the two biological systems which may represent different substrates for the activity of shL1s and with potentially different regulative functions.

At this regard, since the expression of L1s is spatiotemporal-specific (Faulkner et al., 2009), it is reasonable to speculate that different pools of transcribed L1 RNAs may exert different functions in different tissues and developmental stages. Future studies aimed to investigate transcription at a level of the single L1 element may be helpful to characterize and unfold the complexity and diversity of the pools of transcribed L1s in a precise time, cell-type and tissue, and biological functions related to them.

Therefore, in this new version of the manuscript we have addressed these important points in the following ways:

We have added a better description of the differences between the two model systems:

Lines 91-93:

“To further study L1 RNAs role in neuronal differentiation, we isolated cells from E17.5 embryonic cerebral cortex and cultured them for 21 days *in vitro* (div). This is a well-characterized, widely used primary neuronal cell culture system to model later stages of corticogenesis.”

We started the discussion of DEGs from the differences between the two cellular model systems and not from the commonalities:

Lines 183-193:

“The comparison of the effects of L1 RNAs silencing in the two model systems (*in vivo* and *in vitro*), allows some inferences about L1 function at different times of corticogenesis. A substantial quantity of genes (367) showed an opposite pattern: they were upregulated *in vivo* and downregulated *in vitro* ($P = 9.43e-110$). Their GO terms span from synaptic transmission and signalling to ion transport and neural activity (Extended Data Table 6, Extended Data Fig. 5a). This behaviour could be the consequence of the time of harvesting of cortical neurons for *in vitro* differentiation (E17.5), modelling later stages of the neurodevelopmental cascade with respect to the E12.5 stage tested *in vivo*. Furthermore, the time required for proper AAV-mediated expression of shL1s in cultured cells may delay further the consequences on neuronal maturation and cell type composition. Most of the pro-neurogenic effect *in vivo* could be due to the impact of L1 silencing upon proliferation and commitment of progenitor cells, which are much less represented at E17.5.”

We also gave the highest attention to define the type of the model system to which the DEG signature is referred to, every time a specific gene expression profile is quoted:

i. e. in the Abstract: In cortical cultured neurons, L1 RNAs are mainly associated to chromatin and influence the deposition of tri-methylated histone H3 (H3K27me3) marks.

Lines 252-252:

In summary, L1 RNAs silencing in cultured cortical cells was responsible for an overall higher deposition of H3K27me3 repressive mark on genes down-regulated according to RNA-seq experiments.

Importantly, once differences between the two model systems are appropriately described and discussed, it is evident that GO terms enriched by down-regulated and H3K27me3 hypermethylated genes (Extended Data Table 7, Fig. 4g) are related to, and are consistent with, cultured cortical cells isolated from the E17.5 mouse cortex (*in vitro*) (Extended Data Table 3, Fig. 2f, bottom). Because of the different effects produced by L1 RNAs silencing on *in vivo* and *in vitro* systems, it is not therefore surprising that ChIP-seq results on cultured cells were not consistent with those by RNA-seq on developing cortex *in vivo*.

Regarding L1-RT inhibition, we performed the experiment on *in vitro* cultured cortical cells to understand how much of the overall effect produced by L1 RNAs silencing was due to decreased L1 retrotransposition, which can have an impact on the physiology of a cell also by altering gene expression (Blaudin de Thé FX et al. 2018). As discussed in the manuscript, only a small set of genes was dysregulated after pharmacological L1-RT inhibition with AZT and none with 3T3 compared to L1 silencing by shRNAs, and GO terms enriched by those common genes were mostly unrelated to those from L1 silencing *in vitro*. These results strengthen our hypothesis that the main role of L1s in maturing neurons is to contribute to regulation of gene expression by acting as non-coding RNAs. We therefore eliminated the GO analysis from AZT-treated cells since it might negatively contribute to the clarity of the manuscript. Again, for the reasons stated above we cannot compare results of experiments done on *in vitro* cultured neurons (included L1-RT inhibition) with those performed on *in vivo* mouse cortex.

Here are some additional comments, mostly centered around the consistency across experiments:

- 1) *Extended figure 1i was cited in the wrong place in the manuscript.*

Thanks to the reviewer's comment.

We amended this point in the text accordingly (lines 45-50):

“Among post-mitotic neurons, cells expressing the transcription factors Tbr1 and Ctip2, markers of deep-layer neurons, were reduced at E14.5 ($P = 0.0030$ and $P = 0.0025$, respectively; Extended Data Fig. 1g-h). At E18.5, the proportions of Tbr1 and Ctip2 cells were restored, Cux1⁺ upper-layer callosal neurons were more than doubled ($P = 0.0086$), while those expressing the marker Satb2⁺, another upper-layer callosal marker, were similar to controls (Extended Data Fig. 1i-j).”

- 2) *Line 111: “indicated 1129 common genes, of which 215 ($P = 7.71e-33$)”. What is the significance of the overlap of 1129 common genes? Were they consistently upregulated and downregulated in both experiments? It's also unclear what the P value is testing on.*

Thanks to the reviewer's comment because it gives us to opportunity to express with greater clarity these results.

We amended this point in the text accordingly (lines 77-80):

“Comparing the list of differentially expressed genes (DEGs) with those altered upon L1 RNAs silencing in mESCs (9) indicated 215 downregulated genes in common ($P = 5.1e-17$, Fisher's exact test, Jaccard index = 0.1) and 588 shared upregulated genes ($P = 0.059$, Fisher's exact test, Jaccard index = 0.1). The 215 down-regulated genes are shown in Extended Data Table 2.”

3) *Line 143: “3028 common DEGs”. What is the statistical significance of the overlap?*

Thanks for the reviewer's comment.

We amended this point in the text accordingly (lines 115-125):

“The shL1-a dramatically changed cultured cell's transcriptional profile, with 3968 upregulated genes and 3933 downregulated genes when compared to the shCtrl (FDR ≤ 0.05 , limma voom test). A similar observation was made using shL1-b, with 4758 upregulated genes and 2958 downregulated genes when compared to the shCtrl (FDR ≤ 0.05 , limma voom test). The Jaccard similarity coefficient between shL1-a and shL1-b was 0.5 and 0.4 for the upregulated and downregulated gene sets, respectively. The statistical testing of the overlapping genes was significant for the common upregulated genes ($n = 2710$, $P = 0$, Fisher's exact test) and the common downregulated genes ($n = 2038$, $P = 0$, Fisher's exact test). Therefore, we combined the results for the two shL1s and a more stringent filtering was applied by intersecting the consistently dysregulated genes obtained when comparing the two shL1s versus shCtrl and also versus the n.i. cells. This led to 3028 common DEGs (1606 upregulated and 1422 downregulated).”

4) *Line 191: “10.4% of genes altered by shL1-a and -b”. What are the directions (i.e., upregulation and downregulation) and the statistical significance of the overlap?*

Thanks for the reviewer's comment.

We amended this point in the text accordingly (lines 170-182):

“While treatment with 3TC did not significantly alter cells' gene expression profile as measured by RNA-Seq, AZT dysregulated 304 genes (294 down-regulated and 10 up-regulated, FDR ≤ 0.05 , limma voom test; Extended Data Table 5). There was no significant overlap when comparing the 304 genes to those from the shL1-a condition (75 common down-regulated genes and 3 common up-regulated genes, Jaccard index = 0, $P = 0.98$ and $P = 0.65$ respectively, Fisher's exact test) nor to shL1-b ($n = 31$ common down-regulated genes and 5 common up-regulated genes, Jaccard index = 0, $P = 1$ and $P = 0.31$ respectively, Fisher's exact test). There were only 15 down-regulated genes (*C1ql1*, *6030443J06Rik*, *Pcdh15*, *Qk*, *Epn2*, *Zfp521*, *Xylt1*, *Gpm6b*, *Spon1*, *Pdgfra*, *Zcchc24*, *Lhfpl3*, *Sox6*, *Rgcc*, *Asrgl1*) and 2 up-regulated genes (*Lgi2* and *Npas4*) that were consistently affected by the two L1 RNAs silencing and AZT (indicated as consistent at the top of Extended Data Table 5,

Extended Data Fig. 4b-c). These results suggest that L1-RT activity has a limited contribution upon the overall effect on gene expression promoted by L1 RNAs.”

- 5) *Line 303: “FDR < 0.05”. Please add the 95% confidence intervals to Figure 4a to support the statistical significance. If L1-gene regulation is partially through Ezh2 and the histone modification, what is the correlation between gene down-regulation and H3K27Me3 hypermethylation or Ezh2 binding? The consistency between H3K27Me3 hypermethylation and Ezh2 binding as shown in Figure 4b is less significant as that's the known function of Ezh2. Finally, the changes in Ezh2 binding in Figure 4d are hardly discernible.*

Thanks to the reviewer’s comment, we realized that the position of the FDR indication was misleading, since it is referred to the differential analysis shown in panel 4b.

We amended this point in the text accordingly (lines 238-233):

“L1 RNAs silencing in cultured cortical mouse cells was responsible for an overall higher deposition of H3K27me3 repressive mark on the TSS of genes down-regulated according to RNA-seq experiments (Fig. 3e), with 4.641 genomic regions hypermethylated after L1 RNAs silencing compared to control cells (FDR < 0.05, DiffBind test; Extended Data Table 7, Fig. 3f).”

In addition, we now show 95% intervals of confidence in revised Figure 3e (as shaded areas) following the reviewer’s suggestion.

About the correlation between down-regulated genes by RNA-seq and H3K27me3 hypermethylation, or Ezh2 binding, we performed the following analysis:

For H3K27me3, 4.641 genomic regions (FDR < 0.05, DiffBind test) were hypermethylated after L1 RNAs silencing compared to control cells (Extended Data Table 7, Fig. 3f). We tested the significance of the overlap between these 4641 peaks and the 1422 genes consistently downregulated *in vitro* by shL1-a and shL1-b. This overlap was significant (405 regions in common, $P = 0.0009$, enrichPeakOverlap test based on 1000x random sampling of genic regions and present in the UCSC known Gene dataset). As a comparison, the overlap between the 4641 hypermethylated peaks and the 1606 consistently upregulated genes *in vitro* by shL1-a and shL1-b, was not significant (139 regions in common, $P = 0.99$, enrichPeakOverlap test).

We amended this point in the text accordingly (lines 241-244):

“405 regions out of 4641 were significantly enriched with down-regulated genes ($P = 0.0009$, enrichPeakOverlap test; Extended Data Table 3), while the overlap with up-regulated genes was not significant (139 regions, $P = 0.99$, enrichPeakOverlap).”

For Ezh2, 412 regions showed an increased Ezh2 binding (FDR < 0.05, DiffBind test) after L1 RNAs silencing. The overlap between these 412 regions and the 1422 downregulated genes was below significance level (44 regions in common, $P = 0.03$, enrichPeakOverlap test). As a comparison, the overlap between the 412 regions and the 1606 upregulated genes was not significant (19 regions in common, $P = 0.75$, enrichPeakOverlap test).

We amended this point in the text accordingly (lines 241-244):

“L1 RNAs silencing in cultured cortical cells was responsible for a higher binding of Ezh2 in 412 regions (FDR < 0.05, DiffBind test; Extended Data Table 7, Fig. 4f). The majority of Ezh2 bound regions, 261 (62%; $P = 7.6E-11$), overlapped with H3K27me3 hypermethylated sites, suggesting that L1 RNAs knockdown was able to alter, at least for these sequences, both Ezh2 occupancy and H3K27me3 deposition.”

and lines 311-313:

“44 of the regions with an overall higher binding of Ezh2 were significantly enriched with down-regulated genes ($P = 0.03$, enrichPeakOverlap test), while no significant overlap was observed with up-regulated genes ($P = 0.75$, enrichPeakOverlap test).”

6) *Were there no biological replicates in the ChIP-seq analysis? If so, how did the statistics calculated in Figure 4b?*

Thanks to the reviewer’s comment. The ChIP-seq experiments (for both H3K27me3 and Ezh2) were carried out in two independent biological replicates.

We amended this point for H3K27me3 deposition (lines 693-696) in the Fig. 3 legend accordingly:

“e, Metagene plot showing H3K27me3, GFP and input ChIP-seq signals for down-regulated, unchanged and up-regulated genes in mouse cortical cells infected with shCtrl or shL1-a. The plot shows the average signals of $n = 2$ independent biological replicates for each condition with 95% intervals of confidence as shaded areas. f, Volcano plots representing differentially enriched ChIP-seq peaks in shL1-a cortical cells for H3K27me3. X-axis shows the \log_2 (Fold Change). The Y-axis shows the $-\log_{10}$ (FDR). $n = 2$ independent biological replicates.”

And for Ezh2 in the Fig. 4 legend accordingly:

“f, Volcano plots representing differentially enriched ChIP-seq peaks in shL1-a cortical cells for Ezh2. X-axis shows the \log_2 (Fold Change). The Y-axis shows the $-\log_{10}$ (FDR). $n = 2$ independent biological replicates. g, Top GO terms under the biological process category for genes contained in peaks/islands characterized by a significantly higher deposition of Ezh2 and H3K27me3.”

REVIEWERS' COMMENTS

Reviewer #2 (Remarks to the Author):

The authors addressed most of the points that I raised or provided acceptable explanations for most experiments they wish to leave for future studies. I believe that this manuscript will be suitable for publication after the authors will address the following two minor points:

1. The authors still did not demonstrate causality between L1-binding by PRC2 to the regulation of PRC2 (point 4 in my previous reviews). Nevertheless, the authors provided an acceptable explanation for why establishing a mechanism (and causality) would require further development beyond this study and they also moderated some related statements in the main text. Yet, that means that the following statement within the abstract is still unsupported: "L1 RNAs exert some of their regulatory functions by interacting with the Polycomb Repressive Complex 2" (abstract, L24-25). Specifically, the data in the manuscript indicate that PRC2 binds to L1 and, separately, that L1 depletion leads to changes in the ability of PRC2 to bind and modify some target genes, but the authors did not show that L1 exert regulatory functions "by interacting" with PRC2. The authors did well in moderating related unsupported statements within the text but it would be appropriate if they will do so also in the abstract.

2. Fig S6 was modified, but the resolution of some of the panels is too low for reading axes legends (mainly S6g,h).

Reviewer #4 (Remarks to the Author):

I have considered in detail the new changes made by the authors to clarify my previous concerns in regard to the consistency of the measured impacts from L1 KO. There is now sufficient evidence for (1) L1 RNA plays a role in neurogenesis and gliogenesis and (2) likely functions partially through interactions with PRC2 and Suz12. However, I am still concerned that the specific roles of L1 RNA in this process have yet to be fully characterized, likely due to the limitations of the high throughput omic analyses.

Here is another summary of the major experiments to quantify the L1 KO impacts:

- 1) In vivo KO;
- 2) In vitro KO;
- 3) RT inhibition;
- 4) H3K27Me3 ChipSeq;
- 5) Ezh2 ChIPseq

In this new revision, the authors explained well about the difference among the (1) in vivo, (2) in vitro, and (3) RT inhibition analyses, likely due to the developmental timing or the specific regulation mechanism (through RNA, not RT). The authors also showed significant consistency among the biological replicates and the direction of the gene regulations (e.g., higher H3K27Me3, more Ezh2 binding, and less transcription).

Mechanistically, however, there still doesn't seem to be any clear developmental pathways that are consistently regulated as measured across most of these five assays.

A. To compare between 2) in vitro and 4) methylation, the authors said: "Consistent with GO terms enriched by down-regulated genes after L1 RNAs silencing (Extended Data Table 3, Fig. 2 f, bottom), GO analysis of the fraction of these genes contained in hypermethylated regions revealed an enrichment for biological processes related to regulation of neuronal development and for functionalities that include calcium ion concentration, action potential, neuron projection development, synaptic transmission, assembly, plasticity and potentiation (Extended Data Table 7, Fig. 3g)" [[row 274-279]].

However, I failed to see the "regulation of neuronal development" highlighted in the word cloud (Fig. 2f). It only shows up as the 83th most enriched GO category in the extended data Table 3. As a comparison, the "positive regulation of neuronal development" showed up as the 11th most enriched category in the methylation analysis. I am unsure if the (1) vastly different significance and (2) different GO terms can warrant such a strong statement about the consistency of the results.

B. To compare between 2) in vitro, 4) methylation, and 5) Ezh2 binding, the authors said: "Interestingly, genes affected by both increased deposition of Ezh2 and H3K27me3 mark were important for transcriptional regulation of neuronal fate commitment and glial and

oligodendrocyte lineages specification as well as for neuronal cell functionalities, including development of neuronal projections and synaptic processes (Extended Data Table 7, Fig. 4g).” [[row 335 -341]].

And

“ These data suggest that L1 RNAs may influence PRC2-dependent deposition of repressive epigenetic marks on genes crucial for neuronal cell differentiation and activity, a prediction that has been validated with CHIP-seq experiments.” [[row 391-394]].

May I ask why the authors altered the name of the GO category “positive regulation of nervous system development” (as shown in Extended Table 7) to “nervous system development” in the main Figure 3g? In addition, the authors altered the name of the GO category “positive regulation of neuron differentiation” (as shown in Extended Table 7) to “neuron differentiation” in the main Figure 4g. I doubt it is due to the limited space, as the authors used the proper labeling in the first manuscript but changed it in the recent revision.

The critical contradiction here, is that the authors suggested the L1 RNAs play a role in limiting neuronal commitment (row 85, the in vivo assay) and L1 KO promotes neuron differentiation, but in the methylation analysis and Ezh2 binding assays, the authors showed the genes that are positively regulating neuron differentiation are significantly “down-regulated” after L1 KO. By the way, here is the definition of this GO category from AmiGO -- any process that activates or increases the frequency, rate, or extent of neuron differentiation.

I highlighted this issue in my previous review, and I am disappointed to see it remain unanswered and instead changed towards what seems to be a cover-up in the main figures. Despite the difference between the in vitro and the in vivo experiments and the fact that several genes play a role in both positive and negative regulations, the authors should still discuss the seemingly opposite impacts of L1 KO for genes involving neuron differentiation.

Additionally, the authors mentioned adding shaded areas in Figure 3e to represent the confidence intervals, but there are no such areas in the new figure -- it looks exactly like the

old one.

I apologize for further dragging on the reviewing process. However, the exact impacts of L1 RNA in neuron/glia differentiation remain unclear in the current state of the manuscript. I also suggest the authors use full GO category names in the figures, to avoid unnecessary confusion in interpreting the results.

Reviewer's Comments:

Reviewer #2 (Remarks to the Author)

The authors addressed most of the points that I raised or provided acceptable explanations for most experiments they wish to leave for future studies. I believe that this manuscript will be suitable for publication after the authors will address the following two minor points:

1. The authors still did not demonstrate causality between L1-binding by PRC2 to the regulation of PRC2 (point 4 in my previous reviews). Nevertheless, the authors provided an acceptable explanation for why establishing a mechanism (and causality) would require further development beyond this study and they also moderated some related statements in the main text. Yet, that means that the following statement within the abstract is still unsupported: "L1 RNAs exert some of their regulatory functions by interacting with the Polycomb Repressive Complex 2" (abstract, L24-25). Specifically, the data in the manuscript indicate that PRC2 binds to L1 and, separately, that L1 depletion leads to changes in the ability of PRC2 to bind and modify some target genes, but the authors did not show that L1 exert regulatory functions "by interacting" with PRC2. The authors did well in moderating related unsupported statements within the text but it would be appropriate if they will do so also in the abstract.

We thank the Reviewer for her/his comment. We amended the abstract accordingly:

"In cortical cultured neurons, L1 RNAs are mainly associated to chromatin and interact with the Polycomb Repressive Complex 2 (PRC2) protein subunits *enhancer of Zeste homolog 2* (Ezh2) and *suppressor of zeste 12* (Suz12). L1 RNA silencing influences PRC2's ability to bind a portion of its targets and the deposition of tri-methylated histone H3 (H3K27me3) marks."

2. Fig S6 was modified, but the resolution of some of the panels is too low for reading axes legends (mainly S6g,h).

We thank the Reviewer for her/his comment. We modified the Extended Data Fig. 6 making panels g and h 50% larger. Also, we increased the text size of some other elements (i.e. axis legends and labeling). These changes should help the readability of the figure.

Reviewer #4 (Remarks to the Author)

I have considered in detail the new changes made by the authors to clarify my previous concerns in regard to the consistency of the measured impacts from L1 KO. There is now sufficient evidence for (1) L1 RNA plays a role in neurogenesis and gliogenesis and (2) likely functions partially through interactions with PRC2 and Suz12. However, I am still concerned that the specific roles of L1 RNA in this process have yet to be fully characterized, likely due to the limitations of the high throughput omic analyses.

Here is another summary of the major experiments to quantify the L1 KO impacts:

- 1) In vivo KO;
- 2) In vitro KO;
- 3) RT inhibition;
- 4) H3K27Me3 ChipSeq;
- 5) Ezh2 ChIPseq

In this new revision, the authors explained well about the difference among the (1) in vivo, (2) in vitro, and (3) RT inhibition analyses, likely due to the developmental timing or the specific regulation mechanism (through RNA, not RT). The authors also showed significant consistency among the biological replicates and the direction of the gene regulations (e.g., higher H3K27Me3, more Ezh2 binding, and less transcription).

Mechanistically, however, there still doesn't seem to be any clear developmental pathways that are consistently regulated as measured across most of these five assays.

A. To compare between 2) in vitro and 4) methylation, the authors said: "Consistent with GO terms enriched by down-regulated genes after L1 RNAs silencing (Extended Data Table 3, Fig. 2 f, bottom), GO analysis of the fraction of these genes contained in hypermethylated regions revealed an enrichment for biological processes related to regulation of neuronal development and for functionalities that include calcium ion concentration, action potential, neuron projection development, synaptic transmission, assembly, plasticity and potentiation (Extended Data Table 7, Fig. 3g)" [[row 274-279]].

However, I failed to see the "regulation of neuronal development" highlighted in the word cloud (Fig. 2f). It only shows up as the 83th most enriched GO category in the extended data Table 3. As a comparison, the "positive regulation of neuronal development" showed up as the 11th most enriched category in the methylation analysis. I am unsure if the (1) vastly different significance and (2) different GO terms can warrant such a strong statement about the consistency of the results.

We thank the Reviewer for her/his comment that give us the opportunity to improve our article by avoiding too strong statements. We agree that the two lists of GO terms (enriched for all the genes down-regulated by L1 silencing and for those that are down-regulated and contained in peaks/islands

with a significantly higher deposition of H3K27me3) do not perfectly match and show some differences. We therefore reasoned that in our experiments H3K27 hypermethylation targets downregulated genes that are important for processes related to neuronal activity and functionalities supporting the notion that hypermethylation of H3K27 could represent **one, but not the only one**, of the possible mechanisms by which L1 RNAs knockdown could affect gene expression changes and consequent neuronal maturation and functions.

With this reasoning in mind and to meet the Reviewer's comments, we amended the related text accordingly (rows 239-245):

“GO analysis of the fraction of down-regulated genes contained in hypermethylated regions revealed an enrichment for biological processes related to neuronal cells' functions and activities including calcium ion concentration, action potential, cell junctions and membrane rafts assembly, neuron projection development, synaptic transmission, assembly, plasticity and potentiation (Extended Data Table 7, Fig. 3g).

These results suggest that changes in repressive epigenetic marks may account for a portion of the effects on gene expression observed after L1 RNAs silencing in cultured cortical cells.”

B. To compare between 2) in vitro, 4) methylation, and 5) Ezh2 binding, the authors said: “Interestingly, genes affected by both increased deposition of Ezh2 and H3K27me3 mark were important for transcriptional regulation of neuronal fate commitment and glial and oligodendrocyte lineages specification as well as for neuronal cell functionalities, including development of neuronal projections and synaptic processes (Extended Data Table 7, Fig. 4g).” [[row 335 -341]].

And

“These data suggest that L1 RNAs may influence PRC2-dependent deposition of repressive epigenetic marks on genes crucial for neuronal cell differentiation and activity, a prediction that has been validated with ChIP-seq experiments.” [[row 391-394]].

May I ask why the authors altered the name of the GO category “positive regulation of nervous system development” (as shown in Extended Table 7) to “nervous system development” in the main Figure 3g? In addition, the authors altered the name of the GO category “positive regulation of neuron differentiation” (as shown in Extended Table 7) to “neuron differentiation” in the main Figure 4g. I doubt it is due to the limited space, as the authors used the proper labeling in the first manuscript but changed it in the recent revision.

We thank the Reviewer for her/his comment pointing out these inconsistencies. While these modifications were exclusively due to formatting needs, we realize they can generate unwanted confusion in the readers. We apologize for this misunderstanding and amend Fig.3 and Fig.4 accordingly to the Reviewer's suggestion to show the full name of GO categories.

The critical contradiction here, is that the authors suggested the L1 RNAs play a role in limiting neuronal commitment (row 85, the in vivo assay) and L1 KO promotes neuron differentiation, but in

the methylation analysis and Ezh2 binding assays, the authors showed the genes that are positively regulating neuron differentiation are significantly “down-regulated” after L1 KO. By the way, here is the definition of this GO category from AmiGO -- any process that activates or increases the frequency, rate, or extent of neuron differentiation.

I highlighted this issue in my previous review, and I am disappointed to see it remain unanswered and instead changed towards what seems to be a cover-up in the main figures. Despite the difference between the *in vitro* and the *in vivo* experiments and the fact that several genes play a role in both positive and negative regulations, the authors should still discuss the seemingly opposite impacts of L1 KO for genes involving neuron differentiation.

We thank the Reviewer for her/his comment. We apologize if the discussion of this part of the work turned out to be ambiguous and contributed to arise the point that there are contradictory results from different experiments and/or development stages. It's our fault for not properly explain the meaning of our findings.

As properly pointed out by the Reviewer, it is a fact that the two model systems present crucial differences that we discussed in the second revised version of the manuscript at the end of the Paragraph: L1 silencing impairs cortical cell development and maturation *in vitro* (previous rows 187-197). We thank the Reviewer for considering this text appropriate.

Nevertheless, we now realize that we have not adequately discussed a crucial element of L1s biology, their heterogeneity which is probably at the basis of the issue raised by the Reviewer. According to L1Base, the mouse genome contains about 2800 full length intact L1 from different subfamilies and therefore a cell may express hundreds of different independent full length L1 transcripts (<https://doi.org/10.1093/nar/gkw925>, <https://doi.org/10.1101/gr.198301>, <https://doi.org/10.1186/s13100-022-00269-z>, <https://doi.org/10.2741/e537>). Their biological activity can be determined by their location in the genome (i. e. intergenic, intronic, promoter or enhancer-associated) and their length (full length, 5' and 3' truncated). Furthermore, their expression is differentially regulated according to the TFBS content of their promoters. By taking advantage of targeting a large repertory of heterogeneous L1 RNAs, we miss the complexity of the regulated L1 transcriptome and the function associated to single transcripts. Importantly, the cellular repertory of L1s can be different in neuronal progenitor cells and in differentiated neurons (<https://doi.org/10.1101/gr.198301>, <https://doi.org/10.1186/s13100-022-00269-z>, <https://doi.org/10.2741/e537>) while being commonly targeted by ShL1 silencing for their sequence homology.

Therefore, we decided to remove any strong conclusion and move comments on the possible causes about the differences between the two model systems from the Result section (previous rows 187-197) to the Discussion and add a commentary on the heterogeneity of L1s, concluding with the considerations that further studies are needed in order to solve such a complicated issue.

(rows 323-340):

The following text is now present in the Discussion:

“It remains to be determined how L1s can promote both neuronal progenitor proliferation *in vivo* and neuronal differentiation *in vitro*. This is mirrored by the opposite pattern of expression of genes that were upregulated *in vivo* and downregulated *in vitro* and involved in synaptic transmission, signalling, ion transport and neural activity (Extended Data Table 6, Extended Data Fig. 5a). This behaviour could be the consequence of the time of harvesting of cortical neurons for *in vitro* differentiation (E17.5), modelling later stages of the neurodevelopmental cascade with respect to the E12.5 stage tested *in vivo*. Furthermore, the time required for proper AAV-mediated expression of shL1s in cultured cells may delay further the consequences on neuronal maturation and cell type composition. Most of the pro-neurogenic effect *in vivo* could be due to the impact of L1 silencing upon proliferation and commitment of progenitor cells, which are much less represented at E17.5. Importantly, the difference in the effect of L1 silencing may be caused by the heterogeneity of L1 transcripts. Their biological activity may depend on the location in the genome (i. e. intergenic, intronic, promoter or enhancer-associated), length (full length, 5’ and 3’ truncated), protein interaction network and expression, being differentially regulated according to the TFBS content of their promoters. By taking advantage of targeting a large repertory of heterogeneous L1 RNAs, the complexity of the regulated L1 transcriptome and the function associated to single transcripts, are missed. Unveiling the repertory of single L1 transcripts by third generation sequencing technologies, will be instrumental to better define their specific functions in the two model systems.”

Additionally, the authors mentioned adding shaded areas in Figure 3e to represent the confidence intervals, but there are no such areas in the new figure -- it looks exactly like the old one.

We thank the Reviewer for her/his comment. We apologize for the misunderstanding about this point. As we mentioned in the previous response letter, and in the relative figure legend, we amended Fig. 3e adding the 95% confidence intervals as shaded areas. However, variations between replicate samples in the ChIP-seq experiment are so small that is very hard to appreciate the shaded areas in the figure by eye. The Reviewer may notice that they become visible by magnifying the figure.

I apologize for further dragging on the reviewing process. However, the exact impacts of L1 RNA in neuron/glia differentiation remain unclear in the current state of the manuscript. I also suggest the authors use full GO category names in the figures, to avoid unnecessary confusion in interpreting the results.

We are deeply grateful to the Reviewer for her/his time and effort to improve our manuscript. We respectfully think that in this new revised of the manuscript we have addressed all her/his points.